# A harmonic projection and least–squares method for quantifying Kelvin wave activity

Andrew Delman<sup>1</sup>, Janet Sprintall<sup>1</sup>, Julie McClean<sup>1</sup>, and Lynne Talley<sup>1</sup>

<sup>1</sup>Scripps Institution of Oceanography, University of California–San Diego, La Jolla, California, USA.

*Correspondence to:* Corresponding author: Andrew S. Delman, Scripps Institution of Oceanography, University of California–San Diego, La Jolla, California, USA. (adelman@ucsd.edu)

**Abstract.** A new method for isolating the equatorial and coastal Kelvin wave signal from alongtrack satellite altimetry data is presented and applied to sea level anomaly (SLA) observations in the tropical Indian Ocean. The method consists of sequential projections onto the SLA data, starting with meridional or cross-shore Kelvin wave profiles derived from shallow water theory (*y*-projections).

- Next, Fourier basis functions in x-t (along-waveguide distance and time respectively) space with the phase speed ranges of Kelvin and Rossby waves are projected onto the y-projections. After projections in all three dimensions have been carried out, least-squares methods are applied to optimize the non-orthogonal basis function coefficients and minimize the misfit of their along-waveguide forcing and dissipation. Lastly, the westward-propagating (Rossby wave-related) signals are removed,
- generating a Kelvin wave coefficient *K* that represents Kelvin wave activity. Along the Indian Ocean equatorial-coastal waveguide, Hovmöller diagrams of *K* show reduced high–wavenumber noise compared to analogous diagrams of pre-processed sea level anomaly. Results from a Monte Carlo simulation demonstrate that Kelvin wave signals generated *a priori* can be effectively isolated even when superimposed with strong Rossby waves; the signs of all but the weakest Kelvin waves are
- diagnosed correctly in over 90% of cases. When the method is applied to 21 years of satellite observations and the SLA signal associated with K is removed, the large residual in the equatorial SLA signal has a spatial distribution consistent with wind–forced Rossby waves. The equatorial SLA variability in the western part of the basin is poorly correlated with the SLA field associated with K, as the superimposed SLA profile of Rossby waves can distort the true origin locations of Kelvin waves
- in the raw SLA field. Therefore, this method offers improved tracking of Kelvin waves compared to the raw SLA dataset, and may provide the opportunity to study weakly nonlinear aspects of these waves by comparison with linear models.

## 1 Introduction

The quantification of ocean variability associated with equatorial long waves is a topic of great importance for understanding the tropical ocean and its role in climate. Since the advent of satellite altimetry, the surface manifestations of these waves and the wind forcing driving them have been tracked in datasets that now comprise over 20 years of continuous global coverage (e.g., Delcroix et al., 1994; Susanto et al., 1998; Boulanger and Menkes, 1999; Drushka et al., 2010). However, to use these observations to better understand the behavior of these planetary waves and their relation-

30 ship to climate variability, analysis techniques are needed that target the specific signatures of Kelvin and Rossby waves in satellite observations. In particular, the present study was motivated by a need to quantify the relative presence of upwelling vs. downwelling Kelvin waves in the equatorial Indian Ocean and along the coasts of Sumatra and Java, where they are influential in the evolution of Indian Ocean Dipole events (Delman et al., submitted).

A variety of techniques have been employed to quantify equatorial long wave activity from satellite observations; these range from the application of sophisticated data assimilation techniques to meridional projections of sea level anomaly (SLA) data. The data assimilation approaches generally use a linear wave–propagation model, along with Kalman filters (e.g., Miller and Cane, 1989; Fu et al., 1993) or adjoints (e.g., Thacker and Long, 1988; Long and Thacker, 1989a, b) to incorpo-

- rate observations. These techniques are particularly useful for cases where observations are sparse and error-prone, as is often the case for in-situ measurements, and also during the earlier years of satellite observations when spatial resolution was low (e.g., Geosat). As the spatial and temporal coverage of altimeter-derived remote sensing data increased, it was conceivable to estimate Kelvin and Rossby wave activity using solely meridional projections of SLA data, or a combination of SLA
- and current observations. Cane and Sarachik (1981) showed that vectors containing SLA and surface current profiles associated with a given vertical Kelvin wave mode and its associated meridional Rossby wave modes are orthogonal; this orthogonality provided the basis for an equatorial wave decomposition in numerous studies (e.g., Delcroix et al., 1994; Yuan et al., 2004; Yuan and Liu, 2009). Boulanger and Menkes (1995, 1999), BM9599 hereafter, also carried out a decomposition using
- only meridional projections of SLA data that were reasonably consistent with projections derived from in-situ moorings. However, the decomposition of Kelvin and Rossby wave modes based on meridional projections of SLA alone are not orthogonal, and as Yuan et al. (2004) notes, this necessitates the inversion of an ill-conditioned matrix. An alternative approach used complex EOFs of SLA to separate Rossby and Kelvin wave signals in the equatorial Pacific (Susanto et al., 1998);
- one limitation of this method is that complex EOFs by definition constrain the along-waveguide and across-waveguide length scales of the waves, while shallow-water theory only constrains the across-waveguide length scale.

Here we build on the methodology of BM9599, by using the approximate phase speeds as well as cross-waveguide profiles to isolate the Kelvin wave signal. Starting with the SLA meridional

- projections of BM9599, we apply harmonic projections in the along–waveguide direction and in time, followed by a least–squares fit to optimize the non–orthogonal projection coefficients. The result is a Kelvin wave coefficient K that approximates Kelvin wave generation and dissipation along the waveguide, and can be used to track coastal as well as equatorial Kelvin waves. The method as described is focused only on an accurate representation of Kelvin (not Rossby) wave activity, though
- an extension of these techniques might enable a comprehensive decomposition of equatorial waves (as discussed in Section 4). The paper is structured as follows: Section 2 describes the satellite data used, and the harmonic projection and least–squares method that results in the Kelvin wave coefficient K. Section 3 estimates errors associated with the computation of K using a Monte Carlo simulation, and discusses qualitative and quantitative analyses of satellite observations to assess
- how effectively K describes Kelvin wave activity along the Indian Ocean waveguide. Section 4 summarizes the strengths and weaknesses of the method, and considers the possibility of extending the method to quantify Rossby wave activity.

#### 2 Method

# 2.1 Data

- Our methodology quantifies Kelvin wave activity using AVISO Ssalto/Duacs alongtrack SLA data, specifically those from the TOPEX/Poseidon, Jason–1, and Jason–2 satellites. These satellites repeat their orbit over a given track approximately every 10 days, and the data have near–continuous coverage from September 1992 to December 2013. The reason for using alongtrack as opposed to the frequently–used gridded product is the increased spatial resolution in the along–track direc-
- tion, ~1/10° compared to 1/3° for gridded data. One of the advantages of this method is its utility for tracking waves in their transition from equatorial to coastal Kelvin waves. However, quantifying coastal Kelvin waves requires higher spatial resolution, as the baroclinic radius of deformation shrinks from ~400 km at the equator to ~100 km at 10°S. The disadvantage of using the along-track data is the large spacing between tracks in the zonal/alongshore direction (~ 300 km along the coast), but the spacing is still small relative to the along-waveguide length scale of Kelvin waves
  - near the equator, typically >1000 km.

Due to the anisotropy of equatorial-coastal long waves, the offset angle between satellite tracks and meridional cross-sections at the equator is likewise considered to be negligible, and both ascending and descending tracks are used in the analysis. Along the Sumatra and Java coasts, only

90 ascending (SW–NE oriented) tracks are used in Kelvin/Rossby wave projections to best approximate a cross–shore profile. For computational expediency in the least–squares part of the solution, the method was applied to overlapping two–year subsets of the full data record, with each subset overlapping with the next one by a year. The results from each subset were then patched together using a tapered weighted averaging in the overlapping year to create a continuous field of *K* values

for the 21-year period of record (i.e., with 20 subsets patched together). For comparison purposes and to present clear visual snapshots of variability in the Indian Ocean basin, gridded maps of SLA (MSLA) (Ducet et al., 2000) were also used to generate some of the figures in this paper.

## 2.2 Kelvin wave *y*-projections

- The first step in the computation of the Kelvin wave coefficient K is to calculate the projection
  of the SLA data onto a meridional or cross-shore profile of a baroclinic Kelvin wave based on
  linear shallow-water theory (e.g., Gill and Clarke, 1974; McCreary, 1981). We refer to this as the *y*-projection; for an equatorial Kelvin wave it is the same Gaussian profile given in Appendix A2 of
  Boulanger and Menkes (1995), but our analysis also considers coastal Kelvin waves for which the
  wave profile transitions to a decaying exponential away from the equator. For an equatorial-coastal
- Kelvin wave the profile is

$$h_K(y) = h_0 \exp\left[-\frac{\beta\cos\phi}{2c}y^2 \pm \frac{f_0}{c}y\right] \tag{1}$$

where y is the perpendicular distance relative to the equator or the coastline,  $h_0$  is the amplitude (i.e., peak value) of the wave,  $f_0$  is the Coriolis parameter at the latitude where the profile intersects the coast, and  $\phi$  is the angle of orientation of the coast relative to the east-west axis ( $f_0 = 0$  and  $\phi = 0$  for equatorial Kelvin waves). The sign in front of  $(f_0/c)y$  for coastal Kelvin waves is chosen 110 such that the term is always negative. As our focus here is on Indian Ocean Kelvin waves that are

deflected to the south of the equator, y is negative and decreasing away from the coast, and f<sub>0</sub> < 0, so the sign is negative. The value of c for the meridional/cross-shore profiles in this analysis was taken to be 2.5 m s<sup>-1</sup>. This value of c lies between the first-and second-mode baroclinic phase speeds for Kelvin waves in the region, as these two modes account for most Kelvin waves observed in Indian
Ocean SLA (Drushka et al., 2010). However, using c = 2.0 m s<sup>-1</sup> or 3.0 m s<sup>-1</sup> does not produce a substantially different result.

Applied to the altimetry data, the Kelvin wave y-projection is given by

$$K_y = \frac{1}{2} \int_{-r}^{r} \left( h_{\rm SLA} - \overline{h_{\rm SLA}} \right) \frac{h_K - \overline{h_K}}{h_0} \, dy \tag{2}$$

for equatorial Kelvin waves and

$$K_y = \int_{-r}^{0} \left( h_{\rm SLA} - \overline{h_{\rm SLA}} \right) \frac{h_K - \overline{h_K}}{h_0} \, dy \tag{3}$$

for coastal Kelvin waves south of the equator, where  $h_{SLA}$  is the alongtrack altimetry profile, and r120 is the radius for the profile projection. The overbar indicates the meridional  $\bar{a} = 1/(2r) \int_{-r}^{r} a \, dy$  or

140

cross-shore  $\overline{a} = 1/(r) \int_{-r}^{0} a \, dy$  mean (for equatorial and coastal waves respectively) of the profile *a* over the range being integrated. For the *r* value, we used 5° of latitude for equatorial Kelvin waves; *r* was then tapered to a distance equivalent to 3° of latitude along the coasts of Java and Nusa Tenggara to account for the smaller radius of deformation. We note that  $K_y$  is an integrated measure of the sea level displacement; this type of measure is a more consistent indicator of Kelvin wave activity in the equatorial-coastal transition than peak amplitude, since without substantial dissipation, the peak

amplitude of the wave tends to increase poleward as the radius of deformation decreases (Figure 1).

## 2.3 Projection using harmonic basis functions in x and t

After the Kelvin wave y-projections K<sub>y</sub> are computed, the next step in our approach is to project
K<sub>y</sub> onto two-dimensional Fourier basis functions in along-waveguide distance x and time t. One method of separating these components is to assume that a vector b consisting of the alongtrack Kelvin wave projections K<sub>y</sub> can be explained as a linear combination of two-dimensional Fourier basis functions

$$\mathbf{Am} = \mathbf{b} \tag{4}$$

where the columns of A are the basis functions A<sup>cos</sup><sub>m,n</sub> = cos [2π (k<sub>m</sub>x - f<sub>n</sub>t)] and A<sup>sin</sup><sub>m,n</sub> = sin [2π (k<sub>m</sub>x - f<sub>n</sub>t)]
and x and t are along-waveguide distance and time respectively; the Fourier coefficients to be solved for are contained in the vector m.

Basis functions  $A_{m,n}$  that propagate from one side of the basin to the other at constant amplitude are most effective at representing Kelvin waves that similarly propagate across the basin with little change in amplitude. Kelvin waves that are forced and dissipate within the domain, especially with the low wavenumbers common to Kelvin waves, may have some of their energy aliased into westward–propagating signals. To resolve this aliasing issue, we introduce an additional tapering parameter *s* to the basis functions (Figure 2). The basis functions  $A_{m,n,s}$  take the form

$$A_{m,n,s} = \begin{cases} 0, & x \le x_s - \Delta x \\ \left(1 - \frac{x_s - x}{\Delta x}\right) A_{m,n}, & x_s - \Delta x < x < x_s \\ A_{m,n}, & x \ge x_s \end{cases}$$
(5)

The tapering location x<sub>s</sub> is varied at intervals of Δx = 600 km throughout the span of the waveguide, corresponding to the shortest wavelengths resolved along the coastal part of the waveguide (along
the equator the effective Nyquist wavenumber is higher with more satellite tracks used). For s = 1, x<sub>s</sub> = x<sub>W</sub> the western boundary, while for s > 1, x<sub>s</sub> = x<sub>W</sub> + (s − 1)Δx. The forcing and dissipation of a wave within the domain can be expressed as the superposition of basis functions with varying s-values.

Furthermore, to reduce the number of basis functions and make the subsequent least-squares problem less underdetermined, we limit the basis functions to certain phase speed ranges associated with the waves we expect to observe using satellite altimetry. Therefore only basis functions  $A_{m,n,s}$ corresponding to phase speeds  $c_{m,n} = f_n/k_m$  typical of Kelvin waves (1.5 m s<sup>-1</sup>  $\leq c_{m,n} \leq 3.5$  m s<sup>-1</sup>), Rossby waves (-1.2 m s<sup>-1</sup>  $\leq c_{m,n} \leq -0.4$  m s<sup>-1</sup>), and stationary signals ( $k_m = 0$  or  $f_n = 0$ ) are included in  $A_{m,n,s}$ , while the other basis functions are excluded. This phase-speed limitation 155 reduces the number of basis functions to approximately twice the number of  $K_y$  values in b. The

tapered basis functions in  $A_{m,n,s}$  corresponding to the phase speed ranges are projected onto the vector **b** containing the  $K_y$  values. First, a least–squares planar fit to **b** is removed, and then the basis functions are projected onto **b** 

$$\mathbf{m}^{\mathbf{P}} = \mathbf{A}^T \mathbf{b} \tag{6}$$

producing a vector  $\mathbf{m}^{\mathbf{P}}$  of data projections that can be used to solve for the basis function coefficients

160 m of identical size. For the projection values to be unbiased with respect to s, the projections are normalized by the size of the nonzero domain, expressed as a normalizing vector  $\mathbf{n}^{\mathbf{P}}$ , with  $n_{m,n,s}^{P} = (2/N)x_{\max}[x_{\max} - (x_s - 0.5\Delta x)]^{-1}$ , and  $N = \text{length}(\mathbf{b})$ . Accordingly, the normalized vector of basis function projections is

$$\mathbf{m}^{\mathbf{n}\mathbf{P}} = \mathbf{n}^{\mathbf{P}}\mathbf{m}^{\mathbf{P}} = \mathbf{n}^{\mathbf{P}}\mathbf{A}^{T}\mathbf{b}$$
(7)

### 2.4 Least-squares optimization and removal of westward-propagating signals

- After the *x*−*t* projections have been carried out and normalized, basis function coefficients are recovered from the data projections in m<sup>nP</sup> using least–squares methods; the solution is optimized to prevent a poorly-scaled solution with the cancellation of large coefficients in m. In addition to constraining the size of the basis function coefficients m<sup>T</sup>m, we chose to minimize the misfit of the rate of change in data projection values along the waveguide, ∂m<sup>nP</sup>/∂s, in order to constrain high–wavenumber variability within the domain. We also minimize the misfit of the untapered data
- projection values,  $\mathbf{m}^{\mathbf{n}\mathbf{P}}|_{s=1}$ , to the s = 1 basis function coefficients. Hence the vector that we minimize the misfit for is

$$\mathbf{w}^{T}\mathbf{D}\mathbf{m}^{\mathbf{n}\mathbf{P}} = \mathbf{w}^{T} \begin{bmatrix} \mathbf{m}^{\mathbf{n}\mathbf{P}}|_{s=1} \\ \frac{\partial \mathbf{m}^{\mathbf{n}\mathbf{P}}}{\partial s}|_{s>1} \end{bmatrix}$$
(8)

where **D** is the identity operator for s = 1 projections, and the finite difference operator for s > 1projections. The column vector **w** can be used to weight the elements of **Dm**<sup>**n**P</sup> relative to one another. In this case setting **w** to all ones was found to be sufficient, though accuracy may be gained

in certain areas by adjusting the weighting vector; we speculate about one such case in Section 4. With  $\mathbf{w}_n = 1$  for all *n*, we minimize the misfit of  $\mathbf{Dm}^{n\mathbf{P}}$  and the size of  $\mathbf{m}^T\mathbf{m}$  using the cost function

$$L = \left[\mathbf{Dn}^{\mathbf{P}}\mathbf{A}^{T}(\mathbf{Am}) - \mathbf{Dm}^{\mathbf{nP}}\right]^{T} \left[\mathbf{Dn}^{\mathbf{P}}\mathbf{A}^{T}(\mathbf{Am}) - \mathbf{Dm}^{\mathbf{nP}}\right] + \lambda \mathbf{m}^{T}\mathbf{m}$$
(9)

Setting  $\lambda = 0.1$  was found heuristically to produce the most credible reconstructions of the  $K_y$  values in b, while reducing noise at the highest wavenumbers. The coefficient vector m is then given by

$$\mathbf{m} = \left[ \left( \mathbf{D} \mathbf{n}^{\mathbf{P}} \mathbf{A}^{T} \mathbf{A} \right)^{T} \mathbf{D} \mathbf{n}^{\mathbf{P}} \mathbf{A}^{T} \mathbf{A} + \lambda \mathbf{I} \right]^{-1} \mathbf{D} \mathbf{n}^{\mathbf{P}} \mathbf{A}^{T} \mathbf{A} \mathbf{D} \mathbf{m}^{\mathbf{n} \mathbf{P}}$$
(10)

Finally, all coefficients of **m** that correspond to westward–propagating basis functions, i.e.,  $sgn(f_n) = -sgn(k_m)$ , are set to zero. The resulting vector  $\mathbf{m}_{\mathbf{K}}$  is used to reconstruct the  $K_y$  field with the westward–propagating signals removed

$$\mathbf{b}_{\mathbf{K}} = \mathbf{A}\mathbf{m}_{\mathbf{K}} \tag{11}$$

where vector b<sub>K</sub> consists of the Kelvin wave coefficients K as a function of x and t. These computations are carried out for overlapping two-year subsets of the data, which are then merged to create a continuous field of K values for the period Sept. 1992–Dec. 2013 covered by the alongtrack SLA dataset.

## 3 Representations of Kelvin wave activity and error/variance estimates

# 3.1 Comparison of K values with raw SLA

- To demonstrate how well K represents Kelvin wave activity, we present a case study where we compare the raw SLA along the IO equatorial–coastal waveguide during the year 1997 to the  $K_y$  and K values for the same period (Figure 3). The  $K_y$  and K values are calculated from the alongtrack SLA data at points where the satellite tracks cross the waveguide, hence these values are presented as points in Fig. 3b–c, whereas the raw SLA data from the gridded product are contoured in Fig.
- 3a. During the May–July period, the predominant feature in the raw SLA (Fig. 3a) is an eastward– propagating patch of elevated positive SLA, indicative of a downwelling Kelvin wave. However, the Kelvin wave *y*–projection (Fig. 3b) shows that this downwelling wave was both preceded and followed by upwelling Kelvin waves, both of which are much more evident in the Kelvin wave projection (Fig. 3b) than in the raw SLA (Fig. 3a). The *y*–projection still contains a number of
- westward-propagating signals (e.g., Jan.–Feb., and Oct.–Dec.) unrelated to Kelvin waves, and most likely represent Rossby waves flanking the equator. These westward–propagating signals are no

longer visible in the K values for 1997 (Fig. 3c), and the trajectories of alternating upwelling and downwelling Kelvin waves are much more readily apparent.

The values of K can also be re-projected back into two spatial dimensions, to reconstruct the 205 component of the SLA field that is associated with Kelvin wave activity. The reconstructed  $h_K$  is obtained by obtaining the wave amplitude  $h_0$  associated with K

$$h_0(x,t) = \frac{K(x,t)}{\int_{-r}^0 \exp\left[-\frac{\beta\cos\phi}{2c}y^2 - \frac{f_0}{c}y\right]dy}$$
(12)

and substituting into (1). A comparison of the reconstructed  $h_K$  with gridded maps of SLA over a two-month period in 1997 (Figure 4) confirms that the Kelvin wave reconstruction is broadly consistent with the Kelvin wave activity suggested by the gridded SLA field, but also highlights some

- key differences. In late May and early June, an elevated SLA field persists in the eastern equatorial Indian Ocean (Fig. 4b–c), while the reconstructed  $h_K$  indicates that the Kelvin wave activity is changing sign from positive to negative there (Fig. 4g–h). In late June and early July, reconstructed  $h_K$  indicates that upwelling Kelvin wave activity is being generated from approximately 60°E eastward (Fig. 4i–j), while SLA is only substantially depressed east of 90°E (Fig. 4d–e). The discrepancy
- during this latter period is likely accounted for by the influence of downwelling Rossby waves on the SLA field in the central Indian Ocean; these waves have positive SLA maxima near the north and south radii of deformation, and would also elevate SLA (to a lesser extent) at the equator. Therefore, this implies that in the raw SLA field in early July 1997 (Fig. 4e) the upwelling Kelvin wave still present in the central equatorial Indian Ocean would not be apparent; this has potential implications
- for understanding the timing of the upwelling wave and where it was forced.

# 3.2 Monte Carlo-based error estimates

In order to place uncertainty bounds on the method's capacity to remove westward–propagating wave activity from the Kelvin wave estimate, we carried out a Monte Carlo simulation. In this way the method could be applied to propagating waves whose amplitudes and *K* values were known *a priori*.

We created 100 years of randomly–generated basis function coefficients m, using Cholesky factorization (e.g., Gentle, 1998) to construct m fields whose local covariance statistics in wavenumber– frequency (k-f) space resemble values computed from the altimetry data, so that realistic Kelvin and Rossby wave signals could be generated. The m coefficients were adjusted so that their values are partially dependent on the local wave amplitude at the same wavenumber and frequency

$$\mathbf{m}|_{k,f,s} = \left[C_s \sum_{s'=1}^{s-1} \mathbf{m}|_{k,f,s'}\right] + \mathbf{r}$$
(13)

with  $C_s$  the location-dependent adjustment parameter and **r** the Cholesky decomposition-based random component. The variances of the basis functions were also adjusted so that the distributions

of total Kelvin and Rossby wave variance along the waveguide are consistent with the variances obtained from satellite altimetry. Finally, after the artificially–generated eastward– and westward– propagating signals were combined, a small amount of white noise was added to the  $K_y$  fields; the variance of this noise is location–dependent and based on the variance in altimetry observations that could not be explained by either Kelvin or Rossby wave signals.

Once the artificial wave field was constructed, the harmonic projection and least–squares method described in sections 2.3–2.4 was applied to the artificial  $K_y$  field, and the K values derived from the basis function coefficients known *a priori* and deduced from the method were compared. An example

- of this for a given simulated year is shown in Figure 5; the artificially–generated  $K_y$  field contains signals propagating in both directions, though for most of the year the westward–propagating Rossby waves appear to predominate (Fig. 5a). However, a consideration of the Kelvin wave signal  $K_{a \text{ priori}}$  in isolation (Fig. 5b) reveals that in addition to the very strong downwelling wave early in the year, a series of weak and moderate Kelvin waves propagate throughout the year. Many of these weaker waves
- are unidentifiable in the K<sub>y</sub> field with the Rossby wave signals superimposed (Fig. 5a). However, the reconstructed Kelvin wave signal K<sub>reconst</sub>, computed by applying the harmonic projection and least–squares method, recovers most of the weaker Kelvin waves in the K<sub>a priori</sub> signal and reproduces their approximate timing and intensity (Fig. 5c). In the few locations where visible discrepancies between K<sub>a priori</sub> and K<sub>reconst</sub> are present (e.g., the intensities of the Kelvin waves in March–April, east of 90°E), high amplitude westward–propagating signals and/or sharp noisy gradients are present in the

 $K_y$  field.

We now consider the error that is present in the reconstructed signal  $K_{\text{reconst}}$  relative to the original signal  $K_{\text{a priori}}$ , specifically  $\epsilon = K_{\text{reconst}} - K_{\text{a priori}}$ . When the 100-year artificial timeseries is analyzed, it is found that the normalized root-mean-square (RMS) error  $\langle \epsilon^2 \rangle^{1/2} / \langle K_{\text{reconst}}^2 \rangle^{1/2}$  is dependent

- on location along the waveguide as well as whether the fields are spatially– and temporally–filtered (Figure 6a). (Here the angle brackets () denote temporal averaging over the entire 100–year time span of the simulation, but no spatial averaging other than filters applied prior to the error calculation.) The error in recovering the original Kelvin wave signal is highest near the equatorial–coastal transition of the waves, and on the eastern end of the domain; elsewhere it is confined to a fairly narrow
- range. However, the error magnitude also depends on whether a spatial or temporal averaging filter is applied prior to the error calculations. Except for the most error-prone regions, the error associated with unfiltered pointwise values of K is 50% to 60% of the total standard deviation of K. If the K has a spatial moving average (boxcar) filter applied, but temporal averaging is limited to 10 day ranges (the resolution of the original points), the normalized error decreases slightly in most
- locations and is smoother across the waveguide. The error associated with 30–day moving averages of K (a typical timescale for intraseasonal Kelvin waves) decreases more substantially, to 35%–45%in most locations.

285

The probability and cumulative distribution functions associated with errors in K illustrate that errors of the same magnitude as the Kelvin waves themeselves are infrequent when a 30-day mov-

- ing average filter is applied (Fig. 6b). Relative to the total standard deviation in filtered K,  $\sigma_K$ , the magnitude of the errors only exceed  $0.5\sigma_K$  about 10% of the time (either positive or negative), and only exceed  $1\sigma_K$  about 2.5% of the time. In this simulation,  $\sigma_K \approx 1.9 \times 10^4$  m<sup>2</sup>, so the error magnitude is less than  $1 \times 10^4$  m<sup>2</sup> over 90% of the time. If the error is considered relative to the magnitude of the filtered reconstructed  $K_{\text{reconst}}$  at each location and time, the error variance is somewhat
- larger. Even so, with the weakest Kelvin waves  $(|K_{\text{reconst}}| < 0.3\sigma_K)$  excluded, the error will only result in misdiagnosing the sign of most Kelvin waves (i.e.,  $\epsilon/K_{\text{reconst}} > 1$ ) approximately 8.5% of the time. Moreover, the 8.5% incidence decreases further if the threshold for excluding weak Kelvin waves is raised (approximately 5.5% for a  $0.5\sigma_K$  threshold and 2% for a  $1\sigma_K$  threshold); thus sign misdiagnosis using this method is rarely a problem for moderate and strong Kelvin waves.

## 280 3.3 Kelvin wave-related and unrelated SLA characteristics

For a more comprehensive view of the variability encapsulated by the Kelvin wave coefficient K described here, we also consider the reconstructed Kelvin wave–associated SLA field  $h_K$ , in the context of the total SLA field  $h_{SLA}$  observed by satellites over the 21–year period from 1992 to 2013. The SLA at each point in space and time can be considered as the sum of the reconstructed  $h_K$  and a residual  $h_{res}$ 

$$h_{\rm SLA} = h_K + h_{\rm res} \tag{14}$$

with h<sub>res</sub> in theory encompassing contributions to the SLA field from processes unrelated to Kelvin waves, including Rossby and other planetary waves. Figures 7a,b illustrates the variances of h<sub>K</sub> and h<sub>res</sub> respectively, normalized by h<sub>SLA</sub>. The variance ratios of both h<sub>K</sub> (Fig. 7a) and h<sub>res</sub> (Fig. 7b) to h<sub>SLA</sub> exceed 1 along some parts of the waveguide, though this is more commonly the case with h<sub>res</sub>.
(NB: The variance ratios can exceed 1, since the h<sub>K</sub> and h<sub>res</sub> fields are generally not orthogonal. Thus we do not describe the variance ratios as "explained variance" in the traditional sense, but rather compare the variances attributed to Kelvin waves vs. the residual to examine whether the residual

signal is consistent with other phenomena such as Rossby waves.)

Additionally, we compute the correlation between  $h_K$  and the total  $h_{SLA}$  and  $h_{res}$  fields (Figure 8), to consider whether the sign of Kelvin wave activity covaries with that of the total SLA field and the residual. The effective degrees of freedom  $N^*$  for the correlations in Fig. 8 were computed from the decorrelation timescales of  $h_K$ . With 21 years of data, values of  $N^*$  range from approximately 50 to 500 over the spatial domain, with decorrelation timescales ranging from intraseasonal to semiannual. (For  $N^* = 50$ , correlation coefficient magnitudes exceeding 0.23 exceed the 95% confidence

threshold; for  $N^*$  = 500 this threshold is approximately 0.08.) The correlation of  $h_K$  to  $h_{SLA}$  along

the waveguide (Fig. 8a) is strongly positive in the eastern part of the basin and along the coast, but insignificant or negative in the western part of the basin. The correlation of  $h_K$  to  $h_{\rm res}$  is negative over nearly the entire domain, suggesting the tendency of  $h_K$  and  $h_{\rm res}$  to be of opposite sign and explaining how  $h_{\rm res}$  in particular can have a much larger variance than the total SLA field.

- The variance ratios (Fig. 7) and correlations (Fig. 8) suggest different contributions from  $h_K$  and  $h_{\rm res}$  variability in at least four distinct regions along the waveguide. In the western and central parts of the equatorial basin, even the maximum variances of  $h_K$  near the equator are only slightly more than half the variance associated with  $h_{\rm res}$ . In this equatorial region, it is likely that most of  $h_{\rm res}$  can be attributed to Rossby waves; indeed a linear wind–forced model of equatorial waves (Nagura and
- McPhaden, 2010) has shown that in the western part of the basin, Rossby waves are associated with a higher SLA standard deviation than Kelvin waves, even at the equator where Kelvin wave variability peaks. The correlation of  $h_K$  and  $h_{res}$  (Fig. 8) is also strongly negative here, consistent with the expected response of the ocean to a uniform zonal wind forcing, which would generate Kelvin and Rossby waves of opposite sign. In the eastern part of the basin and along the coast of Sumatra, the
- variances of  $h_K$  and  $h_{res}$  are more comparable, though this does not quite resemble the results from the linear forced model (Nagura and McPhaden, 2010) which show a much larger component of SLA due to Kelvin waves than Rossby waves. Near the coast of Java, the variance of  $h_K$  is much larger than that of  $h_{res}$ , suggesting that most of the SLA variability in this area can be attributed to coastal Kelvin waves. Along Nusa Tenggara (NT) the variance of  $h_{res}$  is once again comparable to
- or greater than  $h_K$ ; this may be due in part to the complexity of the islands and bathymetry here. It is also likely that the computation of K does not accurately resolve the splitting and diversion of Kelvin wave energy through Lombok Strait between Java and NT (e.g., Syamsudin et al., 2004; Drushka et al., 2010), since the least–squares fit exhibits a preference for slow tapering of K rather than the abrupt change in wave activity associated with the narrow strait.
- Finally, the lack of a robust correlation between  $h_K$  and  $h_{SLA}$  along the equator in the western part of the basin (Fig. 8) implies that using raw SLA to track Kelvin wave propagation may not accurately represent where waves originate. Namely, SLA crests and troughs that only become clearly apparent in the eastern part of the basin may actually have origins further west; some specific cases of this are evident when comparing the SLA and K values for 1997 (Fig. 3a, c).

#### 330 4 Conclusions

The harmonic projection and least–squares method outlined here produces a measure of Kelvin wave activity that can be applied directly to satellite observations of SLA, not only along the equator, but also along low–latitude coastal waveguides. The method removes the westward–propagating signals (i.e., Rossby wave signals) along such waveguides, and produces K coefficients that represent the time variability of Kelvin wave activity at each location along the waveguide. When filtered to re-

335 time v