# Peer review of "A harmonic projection and least–squares method for quantifying Kelvin wave activity"

_Ocean Science, 2016_

## Referee Comment (RC1) · Anonymous Referee #1 · 10 Apr 2016

This paper proposes a statistical method for extracting the Kelvin wave signal from altimetric data in the Indian ocean both along the equator and along the coast. While I find the topic worth addressing and relevant, I have reservations regarding the method.

1) The paper mentions the study by Boulanger and Menkes (1999) which is a method that provides estimate of the equatorial Kelvin based on the projection of the theoretical meridional wave structures. We wonder why the authors do not test this method to compare with theirs, since the method by BM99 was validated from independent observations (by reconstructing the wave-induced zonal currents and comparing to TAO data). This would provide more confidence in their results if both methods compare to some extent. The method should be at least applied to the equatorial Pacific and compared with estimates from Boulanger and Menkes (1995, 1999).

2) The equatorial Kelvin wave while impinging along the coast of Indonesia could be trapped or reflect as Rossby wave depending on its vertical structure and frequency (Clarke and Shi, 1991). Since the method does not discriminate explicitly the frequencies in the raw SLA, it is not clear to which extend it is able to actually grasp the share of the variability that is trapped along the coast from the one that radiates off-shore as Rossby wave. In fact, the cross-shore scale of the Kelvin wave that is extracted indicates that there is probably a mixture between Kelvin and Rossby waves, and might explain the large residual at some places (Figures 7 and 8). Could the authors comment on that? How are altered the results of the projection when the raw SLA data are filtered in the frequency domain so as to retain intraseasonal frequencies?

3) While in the Pacific the equatorial Kelvin wave structure can be assumed to reduce to the one of the first baroclinic mode, this is certainly not the case for the Indian ocean where the wave structure is more complex and result from the superposition of a number of baroclinic modes. This is a difficulty that is not discussed in the paper and that could be of importance. In fact it is not clear what is the purpose of doing the y-projection considering that this projection does not provide the Kelvin wave coefficient. We would expect that if the method is applied on the raw SLA data at each latitudes (without performing the y-projection), the results of the projection onto the basis functions (i.e. $K(y)$) should have a meridional structure of a Kelvin wave (i.e. a Gaussian curve) from which a phase speed could be inferred. Could the authors verify that? This would be a stringent test that the method does allow extracting the Kelvin wave signal from the raw SLA. It is also expected that the zonal change in the meridional structure reflects the sloping thermocline (i.e. larger value of c to the west).

Other comments:

p. 4, l. 105: please provide a reference for this formula (1)

p. 4, l. 115: this is surprising. Could you illustrate that? I think this is due to the fact that r=5°S. Please explain what is the purpose of the y-projection.

p. 5, l. 137-142: It is not clear if with such a tapering, the basis functions are still a basis? Also this tapering is in fact modelling the dissipation of the waves, it may need to be better justified physically and the sensitivity of the results to the tapering parameters would need to be evaluated.

p. 7, l. 195-200: What is called "The Kelvin wave y-projection" has nothing to do with the wave coefficient associated to the Kelvin wave. The wording is confusing. This quantity does contain variability associated to the Rossby wave, so there is no reason why Figure 3b should exhibit more Kelvin-wave-like properties than the raw SLA. In fact performing the y-projection might be a problem since you mix waves having different propagating characteristics (i.e. phase speed and dissipation rate) so that where the amplitude of Ky accounts for the contribution of the Rossby wave and Kelvin wave in a proportion that does not correspond to the actual ratio expected from the local wind forcing.

p. 8, l. 210: it would be more convincing if a hovmuler of SLA along the coast of Sumatra was provided over an extended period of time.

p. 8, section 3.2: While mathematically correct, I find the test not really relevant. You use the basis functions of propagating modes to construct a surrogate sea level field that you perturb with noise, and then project again on the same basis functions. It is not so surprising that you come up with something you want. However this is not testing if the method is actually able to extract the Kelvin wave signal from altimetric data. It would be more relevant to compare with BM99's method and apply it on data over the equatorial Pacific.

---

## Referee Comment (RC2) · Anonymous Referee #2 · 26 Apr 2016

This manuscript attempts to derive meaningful indices for Kelvin wave activity along the equatorial and coastal waveguide in the Indian Ocean. It is a worthwhile goal, and the authors have certainly invested a great deal of effort in the pursuit, but I regret that I cannot recommend the manuscript for publication. There are too many unsupportable and erroneous mathematical manipulations that don't do what the authors intend them to, and I cannot come up with suggestions for straightforward corrections that would fix the problems. Below I will list problems as they arise in the manuscript, but this list is not necessarily comprehensive. By the point in the manuscript where the list stops, I could no longer believe in the product of the manipulations.

1) Line 58: The authors claim to be building on the methodology of Boulanger and Menkes, 1995, 1999, but this is not true. The problem with deducing equatorial wave amplitudes from only Sea Level Anomalies (SLAs) is that the signals of separate wave modes are not orthogonal (as the authors acknowledge). The individual wave amplitudes could only be deduced if the observed meridional profile of SLA were composed of only a finite number of modes. Boulanger and Menkes made the reasonable approximation that MOST of the SLA profile could be described by a large but finite number of modes (21) and then they derived an approximate solution for the singular matrix associated with projections onto all these modes. This manuscript only attempts a projection onto the lowest Hermite function, which if done properly, would not be able to distinguish between the amplitudes of the Kelvin wave and the first meridional mode long-Rossby wave; both of which can be expected to be present in the SLA signal. Presumably, these can be separated later by separating the eastward from the westward propagating signals, but the ambiguity should be acknowledged up front. It is misleading to call the projection attempted in equation (2) the "Kelvin wave y-projection."

2) The "projections" in (2) and (3) are not really projections, the mathematical forms and the choice of integration limits are puzzling, and the authors provide no justification for their choices. In Fourier analysis it is common to subtract the mean of a data set prior to projecting onto the sines and cosines, but this works because the basis functions all have zero mean. This is not true of the Hermite functions, and it is not true of the exponential profile of the coastal wave. Subtracting the means before integration introduces extra terms that have nothing to do with the desired projection, and limiting the integration limits to 5 degrees latitude ensures that the Kelvin wave structure will not even be orthogonal to structures that it should be orthogonal to.

I'll use a simple idealized situation to illustrate the problems with (2). On an unbounded equatorial $\beta$-plane, the orthonormal Hermite functions, $\psi_n(\hat{y}), \quad n = 0, 1, 2, \cdots$, provide a complete basis. The argument of the Hermite function is the latitude normalized by the deformation radius: $\hat{y} = y/L_e$, with $L_e = \sqrt{c/\beta} \approx 3°$ latitude for $c = 2.5$ m/s. Even in a bounded basin like the Indian Ocean, a truncated set of these functions does a decent job of describing meridional structures while remaining approximately orthonormal. An exception might be the region just south of Sri Lanka, but this is only a small part of the longitudinal span of the basin.

The meridional structure of an equatorial Kelvin wave's SLA is $\psi_0$: the Gaussian part of

the authors' equation (1), with a normalizing factor of $\pi^{-1/4}$ (so that $\int_{-\infty}^{\infty} \psi_0^2 \, d\hat{y} = 1$). The structure of the mode-1 long-Rossby wave's SLA is $(2^{-1/2}\psi_2 + \psi_0)$. Suppose the measured equatorial SLA contains only a Kelvin wave of amplitude $A_K$, a 1st mode long-Rossby wave of amplitude $A_1$ and a background of other variability that is orthogonal to the Kelvin wave: $B(y) = \Sigma_{n=1}^{\infty} b_n \psi_n$. A true projection onto the Kelvin wave structure would be

$$
\begin{aligned}
K_y &= \int_{-\infty}^{\infty} h_{SLA} \, \psi_0 \, d\hat{y} & (1)\\
&= \int_{-\infty}^{\infty} [A_K \psi_0 + A_1(2^{-1/2}\psi_2 + \psi_0) + B] \, \psi_0 \, d\hat{y} & (2)\\
&= (A_K + A_1) \int_{-\infty}^{\infty} \psi_0^2 \, d\hat{y} + 2^{-1/2}A_1 \int_{-\infty}^{\infty} \psi_2 \psi_0 \, d\hat{y} + \Sigma_{n=1}^{\infty} b_n \int_{-\infty}^{\infty} \psi_n \psi_0 \, d\hat{y} & (3)\\
&= A_k + A_1. & (4)
\end{aligned}
$$

This demonstrates the non-orthogonality of the pressure structures of equatorial waves, but at least we're only left with two modes to be sorted out later. If the integration limits are reduced to $\pm 10°$, the results are essentially the same, with only a small error. At the authors' chosen integration limit of 5° latitude, however, the Kelvin wave structure is still almost 25% of it's maximum value, and over this interval it is not even approximately orthogonal to any of the other even Hermite functions. In this case, a proper attempt at a projection would yield:

$$
\begin{aligned}
K_y &= (A_K + A_1) \int_{-5°/Le}^{5°/Le} \psi_0^2 \, d\hat{y} + 2^{-1/2}A_1 \int_{-5°/Le}^{5°/Le} \psi_2 \psi_0 \, d\hat{y} + \Sigma_{n=1}^{\infty} b_n \int_{-5°/Le}^{5°/Le} \psi_n \psi_0 \, d\hat{y} & (5)\\
&= 0.98(A_k + A_1) - 0.06A_1 + 0.08b_2 + 0.06b_4 + \Sigma_{n=6}^{\infty} b_n \int_{-5°/Le}^{5°/Le} \psi_n \psi_0 \, d\hat{y}. & (6)
\end{aligned}
$$

The additional terms may be individually small, but a realistic background would contain a large number of them, and they can add up to a significant number that has nothing to do with the Kelvin wave amplitude, all because the integration was not carried to a latitude where the Kelvin wave is truly insignificant.

An even worse situation arises when the means are subtracted prior to the integration, as in the manuscript's equation (2). I will continue to integrate in the nondimensional coordinate $\hat{y}$ for consistency with the above equations, but note that with the exception of a normalizing constant, the "projection" below is identical to (2) in the manuscript. Integrating over $\hat{y}$, the authors' definition of mean becomes

$$
\overline{a} \equiv \frac{1}{2r/Le} \int_{-r/Le}^{r/Le} a \, d\hat{y}, \tag{7}
$$

and the "projection" in their equation (2) is

$$K_y = \frac{1}{2} \int_{-r/L_e}^{r/L_e} (h_{SLA} - \overline{h_{SLA}})(\psi_0 - \overline{\psi_0}) \, d\hat{y} \tag{8}$$

$$= \frac{1}{2} \int_{-r/L_e}^{r/L_e} h_{SLA} \, \psi_0 \, d\hat{y} - \overline{h_{SLA}} \left(\frac{r}{L_e}\right) \frac{1}{2r/L_E} \int_{-r/L_e}^{r/L_e} \psi_0 \, d\hat{y}$$

$$- \overline{\psi_0} \left(\frac{r}{L_e}\right) \frac{1}{2r/L_E} \int_{-r/L_e}^{r/L_e} h_{SLA} \, d\hat{y} + \overline{h_{SLA}} \, \overline{\psi_0} \frac{1}{2} \int_{-r/L_e}^{r/L_e} d\hat{y} \tag{9}$$

$$= \frac{1}{2} \int_{r/L_e}^{r/L_e} h_{SLA} \, \psi_0 \, d\hat{y} - \frac{r}{L_e} \overline{\psi_0} \, \overline{h_{SLA}}. \tag{10}$$

Using the manuscript's values of $r$ and $c$, the equation is

$$K_y = \frac{1}{2} \int_{-5°/L_e}^{5°/L_e} h_{SLA} \, \psi_0 \, d\hat{y} - 0.85 \, \overline{h_{SLA}}. \tag{11}$$

The first term on the right-hand-side is half of the true projection of the SLA onto the Kelvin wave structure (which includes the mode-1 Rossby wave amplitude), except that it will contain the extraneous terms noted above because the value $r = 5°$ is too small. The second term is a fraction of the mean SLA and has nothing to do with the Kelvin wave amplitude. Furthermore, the factor in front of $\overline{h_{SLA}}$ asymptotes to 0.94 as $r/L_e \to \infty$. This term cannot be removed by choosing larger integration limits. It is the consequence of erroneously subtracting the means of $h_{SLA}$ and $\psi_0$ in (2).

Similar issues arise with the "projection" in (3) onto the coastal Kelvin wave structure, but I won't go into detail. Even without the problem of subtracting the means, the exponential decay of the coastal wave would project onto just about anything. $K_y$ would likely include contributions that have nothing to do with the Kelvin wave or even the Rossby waves represented by their limited set of $x - t$ "basis" functions. To repeat, the "projections" in (2) and (3) are not really projections onto Kelvin wave structures, and they are not reliable measures of either the Kelvin wave or low-mode Rossby wave amplitudes.

3) Lines 124-127: The authors note that the amplitude of a coastal Kelvin wave increases as the wave propagates poleward in the absence of dissipation, and this is correct. They claim, however, that the integral of the Kelvin wave's SLA structure is thus a better measure of the wave, and this is not true. What remains constant with changing latitude is the energy flux, and this is proportional to the integral of the **squared** SLA structure (which obviously includes the squared amplitude).

4) Equation (1) is not quite correct for Kelvin waves propagating meridionally. This may not matter given my next comment, but the authors should at least note the approximation and cite a reference when they present the equation.

5) For a harmonic solution of given frequency, $\omega$, coastal Kelvin waves as described by (1) in the manuscript do not technically exist equatorward of the turning latitudes

$$Y_t = \pm\sqrt{(\omega/\beta)^2 + (c/2\omega)^2}. \tag{12}$$

Within these latitudes, Rossby waves can propagate freely, any meridional motions on an eastern boundary will shed long Rossby waves, and the transfer of energy flux from an equatorial Kelvin wave incident upon an eastern boundary to the outgoing coastal Kelvin waves poleward of the turning latitudes is more complicated than the simple scenario described by (1). For $c = 2.5$ m/s and a period of 45 days, the turning latitudes are at $\pm 7°$, which is about the latitude of the southwestern tip of Java. For periods this long or longer, it is not worth trying to describe a Kelvin wave propagating down the coast of Sumatra from the equator.

The actual Kelvin waves are pulse-like and so contain a spectrum of frequencies, some of which maybe high enough to sustain a true Kelvin wave along the coast of Sumatra, but even if you could describe the alongshore energy flux of the entire pulse as a pseudo-Kelvin wave, its energy flux would be decreasing steadily poleward as energy is shed westward into Rossby modes. This loss of along-shore energy flux would be much different than that produced by dissipation. The tapered pseudo-basis functions would not apply equally well to both situations.

6) The $x - t$ "basis" functions presume that all signals are moving along the coast in one direction or the opposite, and the suite of functions is truncated to waves moving in the Kelvin wave direction for a range of appropriate phase speeds and waves moving in the opposite direction for a range of Rossby wave speeds. If a Kelvin wave could be described along the coast of Sumatra, the Rossby waves would not propagate in the opposite direction from the Kelvin wave. The basis function would be trying to isolate signals propagating northwestward at the Rossby wave's westward phase speed. The Rossby wave's amplitude is embedded in the $K_y$ vector, but the amplitude of the waves shedding westward at one latitude does not match the amplitude of the wave shedding westward at a different latitude in a way described by the basis function.

7) There is often an energetic eddy field south of Java and the Indonesian Islands that is close enough to the coast to be captured by the $K_y$ integration in (3). This is a place where the Kelvin and Rossby waves do propagate in opposite directions, but the manuscript limits the westward propagating basis functions to phase speeds $-1.2$ m/s$< c_{m,n} < -0.4$ m/s. The eddies propagate at -0.15 to -0.2 m/s (Feng and Wijfells, JPO, 2002), so even though their SLA signal will be included in the $K_y$ vector, the basis functions will not be able to filter them out through their propagation characteristics.

8) The "basis" functions are not truly basis functions - they do not constitute a complete set. I would have to see more details of the mathematic involved in the "projections" onto the tapered functions to asses the value of this step.

Unfortunately, at this point I have lost confidence in the authors' understanding of precisely what each of their calculations does. It feels like a series of flawed mathematical manipulations that produces an answer of dubious value. The fact that their figures qualitatively resemble the raw SLA plots is probably due to the robustness of the Kelvin wave in this part of the world, but I can't believe that the final $K$ product has more value than an amplitude derived by simply tracking alongshore time-longitude plots. The

mathematical complexity of the procedure implies a rigor that is not really justified.

The authors have obviously put considerable effort into this procedure and it is possible that with the above comments in mind and a more careful approach, they could rework the procedure into something more meaningful. I can't suggest an obvious way forward, however, so I regret that I cannot recommend the present manuscript for publication.

---

## Editor Comment (EC1) · M. Hecht (Editor) · 27 Apr 2016

Dear Mr. Delman and co-authors,

while Referee #1 asks that your method should be "at least applied to the equatorial Pacific and compared with estimates from Boulanger and Menkes (1995, 1999)", Referee #2 takes this point further, disagreeing with the assertion that you are building on that work, while offering considerable detail in challenging a number of the assumptions made in constructing your method. Referee #1 has also read, and does support, these comments.

Referee 2 acknowledges that "it is possible that with the above comments in mind and a more careful approach, [you] could rework the procedure into something more meaningful". The problem you are addressing is an important one. Accordingly, I will

leave it to you to decide whether to rework your procedure and then attempt a revision; if so, it will be necessary to address the referees' comments to our satisfaction.

Sincerely Yours, –Matthew Hecht
* * *

---

## Author Comment (AC1) · 6 Feb 2017

"A harmonic projection and least-squares method for quantifying Kelvin wave activity"
Response to Reviewer #1

**This paper proposes a statistical method for extracting the Kelvin wave signal from altimetric data in the Indian ocean both along the equator and along the coast. While I find the topic worth addressing and relevant, I have reservations regarding the method.**

General comments: Many thanks to the reviewer for this thoughtful review. Based on feedback received from both reviewers, the manuscript has been reworked so that the method is clearer and easier for others to replicate; the revised method also enables more direct comparisons between our results and the results of other methods that have extracted the Kelvin wave signal from SSH data. Of particular note: since earlier work by Delcroix et al. (1994), Boulanger and Menkes (1995; 1999), and others have simultaneously extracted the signal from Kelvin waves and the lowest meridional mode Rossby waves, we have revised our method to also recover signals representing both Kelvin and Rossby waves. Hence, the title of the article has been changed to "A harmonic projection and least-squares method for quantifying equatorial wave activity". Moreover, the output of our method is now an SLA field associated with each mode (rather than the Kelvin wave coefficient, whose physical significance was not obvious). In the revised manuscript we also exclude the coastal wave part of the decomposition and focus on the more straightforward equatorial region to refine our method; it is hoped that coastal waves can be tracked using a similar method in a future study.

Note about the manuscript: All of the text that has been revised in the article (which includes most of the material) is in italics.

**1) The paper mentions the study by Boulanger and Menkes (1999) which is a method that provides estimate of the equatorial Kelvin based on the projection of the theoretical meridional wave structures. We wonder why the authors do not test this method to compare with theirs, since the method by BM99 was validated from independent observations**
**(by reconstructing the wave-induced zonal currents and comparing to TAO data). This would provide more confidence in their results if both methods compare to some extent. The method should be at least applied to the equatorial Pacific and compared with estimates from Boulanger and Menkes (1995, 1999).**

The reviewer makes a very good point here; in the original manuscript we applied our method only to the Indian Ocean so a direct comparison could not be made between our results and those from Boulanger and Menkes (1999). This has been rectified; in the revised manuscript the method is applied to SLA data in both the equatorial Indian and Pacific Oceans. To facilitate the comparison with Boulanger and Menkes (1999) in particular, Figure 7 shows the Kelvin and mode 1 Rossby waves in the same format and using the same units as in Figure 6 of Boulanger and Menkes (1999). The results are broadly similar, though our method better highlights the higher-frequency (intraseasonal) propagating waves more as described in Section 4 (lines 290-304).

**2) The equatorial Kelvin wave while impinging along the coast of Indonesia could be trapped or reflect as Rossby wave depending on its vertical structure and frequency (Clarke and Shi, 1991). Since the method does not discriminate explicitly the frequencies in the raw SLA, it is not clear to which extend it is able to actually grasp the share of the variability that is trapped along the coast from the one that radiates off-shore as Rossby wave. In fact, the cross-shore scale of the Kelvin wave that is extracted indicates that there is probably a mixture between Kelvin and Rossby waves, and might explain the large residual at some places (Figures 7 and 8). Could the authors comment on that? How are altered the results of the projection when the raw SLA data are filtered in the frequency domain so as to retain intraseasonal frequencies?**

As the reviewer noted, when the Kelvin wave reaches the eastern boundary and starts to propagate poleward along the coast, it can shed Rossby waves. In the earlier manuscript, it was assumed that the loss in Kelvin wave energy that resulted from Rossby wave shedding would be adequately represented using the tapered basis functions, but we agree that the distinction between the Kelvin and Rossby wave signals was likely muddled along coastal Sumatra.

Hence, the reviewer's point is a major reason that we decided to focus on the equatorial oceans in the revised manuscript, in order to validate the method in a context that has been extensively studied using theory and observations. In the revised manuscript, the (low-mode) Rossby wave signals are also explicitly extracted, and both Kelvin and Rossby wave signals are validated using correlations with SLA data and residuals (Section 3).

**3) While in the Pacific the equatorial Kelvin wave structure can be assumed to reduce to the one of the first baroclinic mode, this is certainly not the case for the Indian ocean where the wave structure is more complex and result from the superposition of a number of baroclinic modes. This is a difficulty that is not discussed in the paper and that could be of importance. In fact it is not clear what is the purpose of doing the y-projection considering that this projection does not provide the Kelvin wave coefficient. We would expect that if the method is applied on the raw SLA data at each latitudes (without performing the y-projection), the results of the projection onto the basis functions (i.e. K(y)) should have a meridional structure of a Kelvin wave (i.e. a Gaussian curve) from which a phase speed could be inferred. Could the authors verify that? This would be a stringent test that the method does allow extracting the Kelvin wave signal from the raw SLA. It is also expected that the zonal change in the meridional structure reflects the sloping thermocline (i.e. larger value of c to the west).**

In the earlier manuscript, the projection of the meridional (or cross-shore) structure of the wave was carried out first to obtain the Kelvin wave "y-projection", followed by the projection of the harmonic basis functions to obtain the Kelvin wave coefficient. The separation of these two steps was a consequence of the evolution of our thinking and implementation of the code to compute the Kelvin wave coefficient. However, the method has been revised, so that the projections are carried out in x, y, and t simultaneously. This is accomplished by projecting three-dimensional basis functions of the form given in equation (1) of the revised manuscript.

The new three-dimensional basis functions contain both the meridional structures and the harmonic functions that were used previously.  Projecting these three-dimensional structures allows the waves to be differentiated based on both their meridional structures and approximate phase speeds, in a more efficient way than projecting the meridional structure separately.

Additionally, the correlations of the mode 1 SLA field generated by the method with the original SLA data (Figures 2b,d) allow our assumption that the phase speed c = 2.5 m/s in the equatorial Indian Ocean to be tested.  We expect that the correlation values will be highest where the amplitude of the true mode 1 Rossby wave is largest, and indeed the correlation has its maximum values along the axis of maximum amplitude for a mode 1 Rossby wave with a Kelvin wave phase speed of c = 2.5 m/s.  We are also aware that the second baroclinic mode, while not explicitly resolved in this analysis, makes significant contributions to the SLA field in the equatorial Indian Ocean.  As the text in Section 3 of the revised manuscript now reads:

"The 2.5 m s$^{-1}$ phase speed is consistent with the Argo-based 1st baroclinic mode phase speed estimate by Nagura and McPhaden (2012); according to the estimate by Drushka et al. (2010), also based on Argo data, this value would lie between the 1st and 2nd baroclinic mode Kelvin wave phase speeds (2.8 and 1.8 m s$^{-1}$ respectively). As Drushka et al. (2010) and others have found, the first and second baroclinic modes make substantial contributions to intraseasonal wave activity in the equatorial Indian Ocean, so the 2.5 m s$^{-1}$ phase speed value may represent waves that contain both baroclinic modes." (lines 238-244)

**Other comments:**

**p. 4, l. 105: please provide a reference for this formula (1)**

As the focus of the manuscript is now exclusively on equatorial wave modes, this formula (for a Kelvin wave in the equatorial-coastal transition) is no longer included.

**p. 4, l. 115: this is surprising. Could you illustrate that? I think this is due to the fact that r=5_S. Please explain what is the purpose of the y-projection.**

The fact that the Kelvin wave projections did not depend much on the value of c (at least varying between 2.0 and 3.0 m/s) probably reflects that the Kelvin wave meridional profile is a Gaussian centered along the equator regardless, with slightly different radii.  Specifically, the Kelvin wave profile for c = 2.0 m/s projects highly onto the Kelvin wave profile for c = 3.0 m/s.  This is somewhat less the case for Rossby waves, where the latitude of peak amplitudes actually changes position depending on the value c.

As explained in the response to major comment 3 above, the "y-projection" is no longer a part of the method in the revised manuscript.  Instead, the projections of the meridional structure and the zonally-propagating basis functions are carried out at the same time, by projecting three-dimensional functions of the form given in equation (1) onto the SLA data.

**p. 5, l. 137-142: It is not clear if with such a tapering, the basis functions are still a basis? Also this tapering is in fact modelling the dissipation of the waves, it may need to**

**be better justified physically and the sensitivity of the results to the tapering parameters would need to be evaluated.**

The "basis" functions that are used are not a basis in the traditional sense: a set of functions, orthogonal to one another, that can be summed up (with coefficients) to explain all of the variability in a numerical field. However, even though our "basis" functions are not orthogonal to one another, their application in our method is to achieve a similar goal: to represent the variability present in the field using a set of functions that represent intrinsic modes of variability. To avoid confusion with the traditional mathematical meaning of "basis", these functions have been renamed wave functions.

As for the sensitivity of the results to the tapering parameters, this was tested to some extent by varying the value of w; the parameter w controls the relative weighting in the cost function of the forcing/dissipation associated with the tapers, vs. the cumulative projection values across the basin. The optimization of the parameter w is described in Section 2.3.

**p. 7, l. 195-200: What is called "The Kelvin wave y-projection" has nothing to do with the wave coefficient associated to the Kelvin wave. The wording is confusing. This quantity does contain variability associated to the Rossby wave, so there is no reason why Figure 3b should exhibit more Kelvin-wave-like properties than the raw SLA. In fact performing the y-projection might be a problem since you mix waves having different propagating characteristics (i.e. phase speed and dissipation rate) so that where the amplitude of Ky accounts for the contribution of the Rossby wave and Kelvin wave in a proportion that does not correspond to the actual ratio expected from the local wind forcing.**

It is certainly true that the "Kelvin wave y-projection" as used before contained Rossby as well as Kelvin wave signals; the Rossby wave signals then had to be removed by extracting the westward-propagating signals. As explained in the response to major comment 3, the "y-projection" is no longer carried out prior to the projection of the zonally-propagating functions; instead, three-dimensional wave functions are projected onto the SLA data all at once.

**p. 8, l. 210: it would be more convincing if a hovmuler of SLA along the coast of Sumatra was provided over an extended period of time.**

As the revised analysis no longer includes the coastal part of the waveguide, this comment does not apply.

**p. 8, section 3.2: While mathematically correct, I find the test not really relevant. You use the basis functions of propagating modes to construct a surrogate sea level field that you perturb with noise, and then project again on the same basis functions. It is not so surprising that you come up with something you want. However this is not testing if the method is actually able to extract the Kelvin wave signal from altimetric data. It would be more relevant to compare with BM99's method and apply it on data over the equatorial Pacific.**

The Monte Carlo simulations were an attempt to quantify errors that resulted from the least-squares deconvolution of the non-orthogonal basis functions, and did not include errors that might result from other aspects of the projections. We acknowledge the limits of this approach, and have removed that section in the revised manuscript. Instead, correlations were computed between the SLA associated with each mode and the original SLA data (Figure 2), as well as between the mode SLA and the SLA data residual with SLA from all of the projected

modes removed (Figure 3).  This was done as a means of testing whether the amplitude and phase of the mode contributions to the SLA field are being accurately represented (lines 207-224).

---

## Author Comment (AC2) · 6 Feb 2017

"A harmonic projection and least-squares method for quantifying Kelvin wave activity"
Response to Reviewer #2

**This manuscript attempts to derive meaningful indices for Kelvin wave activity along the equatorial and coastal waveguide in the Indian Ocean. It is a worthwhile goal, and the authors have certainly invested a great deal of effort in the pursuit, but I regret that I cannot recommend the manuscript for publication. There are too many unsupportable and erroneous mathematical manipulations that don't do what the authors intend them to, and I cannot come up with suggestions for straightforward corrections that would fix the problems. Below I will list problems as they arise in the manuscript, but this list is not necessarily comprehensive. By the point in the manuscript where the list stops, I could no longer believe in the product of the manipulations.**

General comments: Many thanks to the reviewer for this detailed, thoughtful review. Based on feedback received from both reviewers, the manuscript has been reworked so that the method is clearer and easier for others to replicate. The method itself has also been changed, such that the projections of the "wave functions" (formerly basis functions) in x, y, and t are carried out simultaneously, rather than carrying out the projections of the structure functions in y before projecting the harmonic functions in x and t. The latitude radius of the projections has also been expanded, from 5° to 15°. We hope that these modifications address some of the major concerns that the reviewer had about the physical significance of each step, and the potential for cross-contamination of Kelvin and Rossby wave signals.

In addition, the revised method also enables more direct comparisons between our results and the results of other methods that have extracted the Kelvin wave signal from SSH data. Of particular note: as in earlier work by Delcroix et al. (1994), Boulanger and Menkes (1995; 1999), and others that extracted the signal from Kelvin waves and the lowest meridional mode Rossby waves, the method in the revised manuscript is used to recover signals representing both Kelvin and Rossby waves. Hence, the title of the article has been changed to "A harmonic projection and least-squares method for quantifying equatorial wave activity". It was also decided to exclude the coastal wave part of the decomposition in the revised manuscript and focus on the more straightforward equatorial region to refine our method; it is hoped that coastal waves can be tracked using a similar method in a future study.

Note about the manuscript: All of the text that has been revised in the article (which includes most of the material) is in italics.

**1) Line 58: The authors claim to be building on the methodology of Boulanger and Menkes, 1995, 1999, but this is not true. The problem with deducing equatorial wave amplitudes from only Sea Level Anomalies (SLAs) is that the signals of separate wave modes are not orthogonal (as the authors acknowledge). The individual wave amplitudes could only be deduced if the observed meridional profile of SLA were composed of only a finite number of modes. Boulanger and Menkes made the reasonable approximation that MOST of the SLA profile could be described by a large but finite number of modes (21) and then they derived an approximate solution for the singular matrix associated with projections onto all these modes. This manuscript only attempts a projection onto the lowest Hermite function, which if done properly, would not be able to distinguish between the amplitudes of the Kelvin wave and the first meridional mode long-Rossby**

wave; both of which can be expected to be present in the SLA signal. Presumably, these can be separated later by separating the eastward from the westward propagating signals, but the ambiguity should be acknowledged up front. It is misleading to call the projection attempted in equation (2) the "Kelvin wave y-projection."

The reviewer is correct about the conceptual framework of our original process: to project the meridional (or cross-shore) Kelvin wave structure onto the SLA data, and then project the meridional projections (the "y-projection") onto propagating harmonic basis functions. Given some of the points raised later in this review, and also in the interest of making the method clearer and more effective, it was decided to eliminate the intermediate step of carrying out the meridional projection in isolation, and instead carry out projections in x, y, and t simultaneously. This is done by projecting onto the SLA data the three-dimensional "wave functions", which are a combination of the meridional structure function in y, the propagating harmonic functions in x and t and the taper functions in x, as shown in equation (1) of the revised manuscript.

2) The "projections" in (2) and (3) are not really projections, the mathematical forms and the choice of integration limits are puzzling, and the authors provide no justification for their choices. In Fourier analysis it is common to subtract the mean of a data set prior to projecting onto the sines and cosines, but this works because the basis functions all have zero mean. This is not true of the Hermite functions, and it is not true of the exponential profile of the coastal wave. Subtracting the means before integration introduces extra terms that have nothing to do with the desired projection, and limiting the integration limits to 5 degrees latitude ensures that the Kelvin wave structure will not even be orthogonal to structures that it should be orthogonal to. I'll use a simple idealized situation to illustrate the problems with (2). On an unbounded equatorial β-plane, the orthonormal Hermite functions, $\psi_n$ y), n = 0, 1, 2,···, provide a (ˆ n complete basis. The argument of the Hermite function is the latitude normalized by the q ∘−y = y/L , with L = c/β ≈−3 latitude for c = 2.5 m/s. Even in deformation radius: ˆ e e a bounded basin like the Indian Ocean, a truncated set of these functions does a decent job of describing meridional structures while remaining approximately orthonormal. An exception might be the region just south of Sri Lanka, but this is only a small part of the longitudinal span of the basin. The meridional structure of an equatorial Kelvin wave's SLA is $\psi_0$ : the Gaussian part of the authors' equation (1), with a normalizing factor of π (so that $\psi_0^2$ dˆ y = 1). The −1/2 structure of the mode-1 long-Rossby wave's SLA is (2 $\psi_2$ + $\psi_0$ ). Suppose the measured 2 0 equatorial SLA contains only a Kelvin wave of amplitude $A_K$ , a 1st mode long-Rossby wave K of amplitude $A_1$ and a background of other variability that is orthogonal to the Kelvin 1 ∞−n=1 n n wave: B(y) = Σ $b_n \psi_n$ . A true projection onto the Kelvin wave structure would be

$$K_y = \int_{-\infty}^{\infty} h_{SLA}\, \psi_0 \, d\hat{y} \tag{1}$$

$$= \int_{-\infty}^{\infty} [A_K\psi_0 + A_1(2^{-1/2}\psi_2 + \psi_0) + B]\, \psi_0 \, d\hat{y} \tag{2}$$

$$= (A_K + A_1)\int_{-\infty}^{\infty} \psi_0^2 \, d\hat{y} + 2^{-1/2}A_1\int_{-\infty}^{\infty} \psi_2\psi_0 \, d\hat{y} + \Sigma_{n=1}^{\infty} b_n \int_{-\infty}^{\infty} \psi_n\psi_0 \, d\hat{y} \tag{3}$$

$$= A_k + A_1. \tag{4}$$

This demonstrates the non-orthogonality of the pressure structures of equatorial waves, but at least we're only left with two modes to be sorted out later. If the integration limits ∘−are reduced to ±10 , the results are essentially the same, with only a small error. At the ∘−authors' chosen integration limit of 5 latitude, however, the Kelvin wave structure is still almost 25% of it's maximum value, and over this interval it is not even approximately orthogonal to any of the other even Hermite functions. In this case, a proper attempt at a projection would yield:

$$K_y = (A_K + A_1) \int_{-5°/Le}^{5°/Le} \psi_0^2 \, d\hat{y} + 2^{-1/2} A_1 \int_{-5°/Le}^{5°/Le} \psi_2 \psi_0 \, d\hat{y} + \Sigma_{n=1}^{\infty} b_n \int_{-5°/Le}^{5°/Le} \psi_n \psi_0 \, d\hat{y} \quad (5)$$

$$= 0.98(A_k + A_1) - 0.06A_1 + 0.08b_2 + 0.06b_4 + \Sigma_{n=6}^{\infty} b_n \int_{-5°/Le}^{5°/Le} \psi_n \psi_0 \, d\hat{y}. \quad (6)$$

The additional terms may be individually small, but a realistic background would contain a large number of them, and they can add up to a significant number that has nothing to do with the Kelvin wave amplitude, all because the integration was not carried to a latitude where the Kelvin wave is truly insignificant. An even worse situation arises when the means are subtracted prior to the integration, as in the manuscript's equation (2). I will continue to integrate in the nondimensional coordinate ˆy for consistency with the above equations, but note that with the exception of a normalizing constant, the "projection" below is identical to (2) in the manuscript. Integrating over ˆy, the authors' definition of mean becomes

$$\bar{a} \equiv \frac{1}{2r/Le} \int_{-r/Le}^{r/Le} a \, d\hat{y}, \quad (7)$$

and the "projection" in their equation (2) is

$$K_y = \frac{1}{2} \int_{-r/L_e}^{r/L_e} (h_{SLA} - \overline{h_{SLA}})(\psi_0 - \overline{\psi_0}) \, d\hat{y} \quad (8)$$

$$= \frac{1}{2} \int_{-r/L_e}^{r/L_e} h_{SLA} \psi_0 \, d\hat{y} - \overline{h_{SLA}} \left(\frac{r}{L_e}\right) \frac{1}{2r/L_E} \int_{-r/L_e}^{r/L_e} \psi_0 \, d\hat{y}$$

$$- \overline{\psi_0} \left(\frac{r}{L_e}\right) \frac{1}{2r/L_E} \int_{-r/L_e}^{r/L_e} h_{SLA} \, d\hat{y} + \overline{h_{SLA}} \, \overline{\psi_0} \frac{1}{2} \int_{-r/L_e}^{r/L_e} d\hat{y} \quad (9)$$

$$= \frac{1}{2} \int_{r/L_e}^{r/L_e} h_{SLA} \psi_0 \, d\hat{y} - \frac{r}{L_e} \overline{\psi_0} \, \overline{h_{SLA}}. \quad (10)$$

Using the manuscript's values of r and c, the equation is

$$K_y = \frac{1}{2} \int_{-5°/L_e}^{5°/L_e} h_{SLA} \psi_0 \, d\hat{y} - 0.85 \, \overline{h_{SLA}}. \quad (11)$$

The first term on the right-hand-side is half of the true projection of the SLA onto the Kelvin wave structure (which includes the mode-1 Rossby wave amplitude), except that it ∘−will contain the extraneous terms noted above because the value r = 5 is too small. The second term is a fraction of the mean SLA and has nothing to do with the Kelvin wave. Furthermore, the factor in front of h SLA e asymptotes to 0.94 as r/L →−∞. amplitude.

**This term cannot be removed by choosing larger integration limits. It is the consequence of erroneously subtracting the means of h and ψ0 in (2).**

**Similar issues arise with the "projection" in (3) onto the coastal Kelvin wave structure, but I won't go into detail. Even without the problem of subtracting the means, the exponential decay of the coastal wave would project onto just about anything. K would y likely include contributions that have nothing to do with the Kelvin wave or even the Rossby waves represented by their limited set of x −−t "basis" functions. To repeat, the "projections" in (2) and (3) are not really projections onto Kelvin wave structures, and they are not reliable measures of either the Kelvin wave or low-mode Rossby wave amplitudes.**

To paraphrase the core issues with this comment, it seems that the reviewer was concerned that (1) the original meridional/cross-shore structure functions being projected have nonzero means, (2) the integration limits are insufficient to distinguish between Kelvin and Rossby wave meridional structures, and (3) removing the means from the basis functions and the SLA fields introduces a spurious term (proportional to the product of the mean values of the Kelvin wave structure and the SLA).  Issue (1) is why the means of the structure functions are removed prior to the projection being carried out (lines 80-83 in the revised manuscript).  It is the meridional shape of the structure function (not the absolute SLA values) that we are trying to recover from the data; if the mean were not removed, then the projection would yield a term that is proportional to the mean value of the SLA along that cross-section.

Issue (2) is a very valid point; the integration limits of 5°S and 5°N were initially chosen to reduce the impact from off-equatorial waves and eddies, as well as to avoid the landmasses of India and Sri Lanka.  However, we acknowledge that the benefits (and arguably the necessity) of expanding the integration range outweigh the earlier concerns about off-equatorial contamination of the signal.  Hence the integration limits have been expanded in the revised manuscript:

"Each wave function is projected onto the SLA data between latitudes 15°S and 15°N, to resolve the meridional structures of equatorial Kelvin waves and the first 5 meridional Rossby wave modes." (lines 100-102)

Additionally, we think the risks from off-equatorial contamination are mitigated by the modification to our method, in which we are projecting the meridional structure and zonal phase propagation simultaneously.  This is because off-equatorial waves propagate more slowly than the lowest equatorial wave modes, and because projecting our wave functions across the basin in x, y, and t reduces the impact from landforms in one small part of the basin.

Issue (3) may result from a confusion about what we are trying to represent.  For example, if the meridional structure of an equatorial Kelvin wave were projected onto an SLA field with essentially infinite integration limits in y, then it would not be necessary to subtract the mean of the meridional structure; the mean would be essentially zero anyway, since the peak in the structure at the equator would be outweighed by its near-zero values on the flanks.  However, in the real ocean we need to project with finite integration limits.  In this case, the first term on the right-hand side of the reviewer's equation (11) does not represent the "true" projection value; it is a weighted mean of the SLA value along the meridional transect.

Our goal is to obtain the same projection value for a Kelvin wave of amplitude 10 cm, regardless of whether the background SLA field it is superimposed on has a mean value of 0 cm or 50 cm. To achieve this, either the structure function or the SLA field (or both) must have the mean removed prior to the projection. Which of these have their mean removed does not make a difference to the value obtained from a projection the structure function onto a 10 cm-amplitude Kelvin wave (due to the cancellation of terms as shown in the reviewer's equations 9 and 10). However, it does make a couple of other important differences. By removing the meridional means of the SLA data, zonal variations in the background SLA values are minimized, which is desirable for equatorial long waves defined dynamically by their meridional pressure and current variations. Removing the means of the structure functions makes the structure functions more orthogonal to one another, and thus better conditions the matrix that must ultimately be inverted. So it is considered desirable to remove the means in both the structure functions and the data. Moreover, the meridional *trends* in both the structure functions and data were also removed; if the integration limits were not expansive enough this would have adverse consequences for the projection of anti-symmetric Rossby wave modes. However, the integration limits of 15° on either side of the equator include almost all of the SLA variability of the first 5 Rossby wave modes, and hence removing these trends excludes the effect of large-scale steric height gradients that we are uninterested in.

**3) Lines 124-127: The authors note that the amplitude of a coastal Kelvin wave increases as the wave propagates poleward in the absence of dissipation, and this is correct. They claim, however, that the integral of the Kelvin wave's SLA structure is thus a better measure of the wave, and this is not true. What remains constant with changing latitude is the energy flux, and this is proportional to the integral of the squared SLA structure (which obviously includes the squared amplitude).**

As the focus of the revised manuscript is now solely on equatorial long waves, this comment is no longer relevant. However, if an attempt is made in the future to apply this method again to coastal waves, we would certainly consider defining the wave functions so that they conserve energy (rather than the integral of the SLA structure) as they propagate.

**4) Equation (1) is not quite correct for Kelvin waves propagating meridionally. This may not matter given my next comment, but the authors should at least note the approximation and cite a reference when they present the equation.**

Again, as the revised manuscript focuses on equatorial long waves, this comment is no longer relevant. We acknowledge that the structure of propagating coastal Kelvin waves is more complex than that indicated by the earlier paper's equation 1 (see also the response to the next comment).

**5) For a harmonic solution of given frequency, ω, coastal Kelvin waves as described by (1) in the manuscript do not technically exist equatorward of the turning latitudes q 2 2 Y = ±−(ω/β) + (c/2ω) . (12) t 3 Within these latitudes, Rossby waves can propagate freely, any meridional motions on an eastern boundary will shed long Rossby waves, and the transfer of energy flux from an equatorial Kelvin wave incident upon an eastern boundary to the outgoing coastal Kelvin waves poleward of the turning latitudes is more complicated than the simple scenario described by (1). For c = 2.5 m/s and a period of 45 days, the turning latitudes are at ◦−±7 , which is about the latitude of the southwestern tip**

of Java. For periods this long or longer, it is not worth trying to describe a Kelvin wave propagating down the coast of Sumatra from the equator. The actual Kelvin waves are pulse-like and so contain a spectrum of frequencies, some of which maybe high enough to sustain a true Kelvin wave along the coast of Sumatra, but even if you could describe the alongshore energy flux of the entire pulse as a pseudo-Kelvin wave, its energy flux would be decreasing steadily poleward as energy is shed westward into Rossby modes. This loss of along-shore energy flux would be much different than that produced by dissipation. The tapered pseudo-basis functions would not apply equally well to both situations.

The complicated dynamics that occur when a Kelvin wave reaches an eastern boundary, resulting in both the reflection of Rossby waves and coastal Kelvin waves, was not fully considered in the earlier version of this article.  This issue likely explained why there was a deterioration in correlation between Kelvin wave coefficient and SLA values along the coast of Sumatra.  In the revised manuscript focused on equatorial waves, we confine ourselves to considering wave reflection by comparing the incoming Kelvin wave amplitudes with the outgoing Rossby wave amplitudes (e.g., Figures 5-7).  In a potential future application of this method to coastal waves, the frequency dependence of the behavior of Kelvin waves impinging on the eastern boundary could be accounted for: either directly by making the meridional/cross-shore structure a function of frequency, or indirectly by tuning the cost function so that the $\Lambda$ diagonal matrix values are a function of the frequency of the associated wave functions.

6) The x − t "basis" functions presume that all signals are moving along the coast in one direction or the opposite, and the suite of functions is truncated to waves moving in the Kelvin wave direction for a range of appropriate phase speeds and waves moving in the opposite direction for a range of Rossby wave speeds. If a Kelvin wave could be described along the coast of Sumatra, the Rossby waves would not propagate in the opposite direction from the Kelvin wave. The basis function would be trying to isolate signals propagating northwestward at the Rossby wave's westward phase speed. The Rossby wave's amplitude is embedded in the K vector, but the amplitude of the waves shedding westward at one y latitude does not match the amplitude of the wave shedding westward at a different latitude in a way described by the basis function.

This is a good point, which will have to be considered should this method be applied in the future to coastal waves.  Using three-dimensional wave functions (as we have done in the revised manuscript) allows much more flexibility in what types of signals can be projected and extracted from the SLA data.  Thus, perhaps a future application could resolve the shedding of Rossby waves propagating zonally at different latitudes, as distinct from the Kelvin waves propagating obliquely meridionally down the coast.

7) There is often an energetic eddy field south of Java and the Indonesian Islands that is close enough to the coast to be captured by the K integration in (3). This is a place where y the Kelvin and Rossby waves do propagate in opposite directions, but the manuscript limits the westward propagating basis functions to phase speeds −1.2m/s< c < −0.4 m/s. m,n The eddies propagate at -0.15 to -0.2 m/s (Feng and Wijfells, JPO, 2002), so even

**though their SLA signal will be included in the K vector, the basis functions will not be able to y filter them out through their propagation characteristics.**

Similarly to the response given to comment 6, this issue could be resolved in the future by projecting wave functions that are bolus-like in the shape of eddies (and propagating in the appropriate latitude and speed ranges), and then removing this signal. There may be more efficient ways to extract the eddy signal though (perhaps by filtering out signals in this latitude band that have short zonal wavelengths).

**8) The "basis" functions are not truly basis functions - they do not constitute a complete set. I would have to see more details of the mathematic involved in the "projections" onto the tapered functions to asses the value of this step.**

The "basis" functions that are used are not a basis in the traditional mathematical sense: a set of functions, orthogonal to one another, that can be summed up (with coefficients) to explain all of the variability in a numerical field. However, even though our "basis" functions are not orthogonal to one another, their application in our method is to achieve a similar goal: to represent as much variability as possible in the field using a set of functions that represent intrinsic modes of variability. To avoid confusion with the mathematical meaning of "basis", these functions have been renamed "wave functions".

It is not clear what aspect of the mathematics the reviewer would need to see to assess the value of the tapered functions. We have found in the revised manuscript that by adjusting the relative weighting $w$ of the misfit to the "taper difference" (the change in the value of the projections as a function of taper location) that we can optimize the correlation values of the mode SLA to the SLA data (Figure 2), and minimize the correlation to the SLA residual (Figure 3). But perhaps the reviewer is seeking a clearer assessment of the improved results that tapered wave functions provide compared to non-tapered wave functions? This would be an interesting analysis, but is beyond the scope of our manuscript at present.

**Unfortunately, at this point I have lost confidence in the authors' understanding of precisely what each of their calculations does. It feels like a series of flawed mathematical manipulations that produces an answer of dubious value. The fact that their figures qualitatively resemble the raw SLA plots is probably due to the robustness of the Kelvin wave in this part of the world, but I can't believe that the final K product has more value than an amplitude derived by simply tracking alongshore time-longitude plots. The mathematical complexity of the procedure implies a rigor that is not really justified. The authors have obviously put considerable effort into this procedure and it is possible that with the above comments in mind and a more careful approach, they could rework the procedure into something more meaningful. I can't suggest an obvious way forward, however, so I regret that I cannot recommend the present manuscript for publication.**

We understand the reviewer's earlier reservations about this method, particularly considering the complex wave structures that are present at the eastern boundary as Kelvin waves impinge on it. The reviewer points out that the resemblance between the Kelvin wave

coefficient in the earlier paper and the raw SLA is likely due to the robustness of the Kelvin wave.  This may indeed have played a role, since the correlations between the Kelvin wave coefficient and the SLA data were highest where the Kelvin wave signal was strongest (in the eastern equatorial Indian Ocean and along the Java coastline), and significantly weaker in some other areas.  Our original goal was to recover a signal that has both the cross-wave structure and the propagation characteristics of a Kelvin wave: we still consider this goal both important and achievable.  The modifications made to our method have improved our ability to parse equatorial wave signals, by expanding the integration limits of the projections, and by projecting the meridional structures and zonal propagation characteristics of the waves simultaneously.  We also focus on the equatorial region in the revised manuscript, deferring the challenges of the coastal wave regime so that the method can be refined by application in the extensively-studied equatorial band.  Lastly, we apply our method to specifically extract the signals of the first 5 meridional Rossby wave modes, as well as Kelvin waves.  This is done to facilitate comparisons to earlier decompositions of equatorial wave activity, especially Boulanger and Menkes (1995; 1999) who also used only SSH data to extract wave amplitudes.  We hope that the reviewer finds these modifications sufficient to reconsider their recommendation.

---

## Author Comment (AC3) · 6 Feb 2017

Manuscript prepared for Ocean Sci.
with version 2015/04/24 7.83 Copernicus papers of the LATEX class copernicus.cls.
Date: 6 February 2017

**A harmonic projection and least–squares method for quantifying *equatorial* wave activity**

Andrew Delman[1], Janet Sprintall[1], Julie McClean[1], and Lynne Talley[1]

[1]Scripps Institution of Oceanography, University of California–San Diego, La Jolla, California, USA.

*Correspondence to:* Corresponding author: Andrew S. Delman, now at the Jet Propulsion Laboratory, Pasadena, California, USA. (Andrew.S.Delman@jpl.nasa.gov)

**Abstract.** A new method for isolating *the sea surface height signals associated with equatorial Kelvin and Rossby waves is* presented and applied to *altimetric* sea level anomaly (SLA) observations in the tropical Indian *and Pacific* oceans. The method *projects wave functions representing propagating equatorial wave modes* onto the SLA data. *Each wave function is three-dimensional: the product of a meridional profile for a given wave mode derived from shallow-water theory, and a harmonic function that propagates zonally within a phase speed range close to that of the wave mode. Moreover, the wave functions are tapered within the zonal domain, to approximate the forcing and dissipation of waves within the domain, and to minimize aliasing of waves that only propagate across part of the ocean basin.* After projections in all three dimensions have been carried out, least-squares methods are applied to *recover* the non-orthogonal wave function coefficients and minimize the misfit of their along-waveguide forcing and dissipation. *The result of these calculations are mode-associated SLA fields associated with Kelvin waves and with the first 5 meridional Rossby wave modes, which can be used as a proxy for the waves' amplitude.*

*The mode SLA field results are validated by correlation with the original SLA data over a 23-year period, as well as by correlation with the residual SLA field from removing the mode SLA. The spatial distribution of the 1st meridional mode Rossby wave correlations also confirms our choices for the value of c, the (1st baroclinic) Kelvin wave phase speed. Compared to earlier methods that used only the meridional structure to decompose equatorial wave modes, the mode SLA clarifies the signals from freely-propagating intraseasonal waves, such as those that are forced by MJO-related winds. As this method of decomposition favors propagating waves but does not constrain their phase speed to a specific value, the mode SLA* may provide the opportunity to study weakly nonlinear aspects of these waves by comparison with linear models.

**1 Introduction**

The quantification of ocean variability associated with equatorial long waves is a topic of great importance for understanding the tropical ocean and its role in climate. Since the advent of satellite

altimetry, the surface manifestations of these waves and the wind forcing driving them have been tracked in datasets that now comprise over 20 years of continuous global coverage (e.g., Delcroix et al., 1994; Susanto et al., 1998; Boulanger and Menkes, 1999; Drushka et al., 2010). However, to use these observations to better understand the behavior of these planetary waves and their relation-

30 ship to climate variability, analysis techniques are needed that target the specific signatures of Kelvin and Rossby waves in satellite observations.

A variety of techniques have been employed to quantify equatorial long wave activity from satellite observations; these range from the application of sophisticated data assimilation techniques to meridional projections of sea level anomaly (SLA) data. The data assimilation approaches generally

35 use a linear wave-propagation model, along with Kalman filters (e.g., Miller and Cane, 1989; Fu et al., 1993) or adjoints (e.g., Thacker and Long, 1988; Long and Thacker, 1989a, b) to incorporate observations. These techniques are particularly useful for cases where observations are sparse and error–prone, as is often the case for in-situ measurements, and also during the earlier years of satellite observations when spatial resolution was low (e.g., Geosat). As the spatial and temporal

40 coverage of altimeter-derived remote sensing data increased, it was conceivable to estimate Kelvin and Rossby wave activity using solely meridional projections of SLA data, or a combination of SLA and current observations. Cane and Sarachik (1981) showed that vectors containing SLA and surface current profiles associated with a given vertical Kelvin wave mode and its associated meridional Rossby wave modes are orthogonal; this orthogonality provided the basis for an equatorial wave de-

45 composition in numerous studies (e.g., Delcroix et al., 1994; Yuan et al., 2004; Yuan and Liu, 2009). Boulanger and Menkes (1995, 1999), BM9599 hereafter, also carried out a decomposition using only meridional projections of SLA data that were reasonably consistent with projections derived from in-situ moorings. However, the decomposition of Kelvin and Rossby wave modes based on meridional projections of SLA alone are not orthogonal, and as Yuan et al. (2004) notes, this ne-

50 cessitates the inversion of an ill-conditioned matrix. An alternative approach used complex EOFs of SLA to separate Rossby and Kelvin wave signals in the equatorial Pacific (Susanto et al., 1998); one limitation of this method is that complex EOFs by definition constrain the along-waveguide and across-waveguide length scales of the waves, while shallow-water theory only constrains the across-waveguide length scale.

55 Here we *outline a method that projects three-dimensional propagating wave functions on SLA values, to estimate the components of equatorial wave modes in the SLA field. In contrast with the work of BM9599 which projected one-dimensional meridional profiles onto SLA, our projection method simultaneously uses the meridional profile and the phase speed of each wave mode to isolate its SLA signal. The method requires only the SLA data as input, and the resulting mode SLA highlights*

60 *intraseasonal propagating features.* The paper is structured as follows: Section 2 describes the data and the harmonic projection and least-squares method *used to obtain the mode SLA field*. Section 3 *assesses the spatial distribution of the relationship between the mode SLA computed and the original*

*SLA field, examining where inaccuracies in the estimation of mode amplitudes may exist. Section 4*
*illustrates characteristics of the mode SLA fields in the two ocean basins, and compares the ampli-*
65   *tudes to those obtained by Boulanger and Menkes (1999).* Section 5 summarizes the strengths and
weaknesses of the method, and considers *possible applications and adjustments to the method in*
*future studies*.

**2  Method**

**2.1  Data**

70   *I*n this study, our methodology *is applied to* AVISO Ssalto/Duacs *gridded maps of absolute dynamic*
*topography (MADT), which are available at 1/4° spatial resolution and daily temporal resolution*
*from the Centre National d'Études Spatiales (Ducet et al., 2000). The analysis discussed here uses*
*the delayed-time, merged product which is produced from all the available altimetry data starting in*
*1993; we make use of the data from January 1993 to December 2015. The resolution of this dataset is*
75   *more than sufficient to track the equatorial long waves targeted in this study, whose maxima/minima*
*generally have meridional scales >1°, zonal scales >5°, and monthly or longer temporal scales (e.g.,*
*Boulanger and Menkes, 1999; Nagura and McPhaden, 2012). However, we note that the projections*
*as described in Section 2.2 could be applied to a non-gridded (e.g., alongtrack) dataset as well,*
*provided the data have sufficient spatiotemporal resolution and coverage.*

80   *Before the projections are applied, the absolute dynamic topography data have their meridional*
*mean and trend between 15°S and 15°N removed at each longitude and time, to obtain the detrended*
*SLA data values $\eta_d$; this excludes signals from large-scale meridional pressure gradients that are*
*unrelated to equatorial wave activity. The SLA data are then band-passed to create a low-passed*
*and a high-passed SLA field; the projections and least-squares deconvolution are applied to the low-*
85   *passed and high-passed fields separately. The lower-frequency wave functions are projected onto*
*the full 23-year low-passed SLA field before the least-squares deconvolution is carried out, while*
*the higher-frequency intraseasonal wave functions are projected onto the high-passed SLA field and*
*deconvolved in overlapping 2-year segments (1993–1994, 1994–1995, 1995–1996, etc.). At the end*
*of the analysis, the low- and high-passed mode SLA components are summed together, with the*
90   *results from the high-frequency 2-year segments combined by tapering the overlapping segments*
*together.*

**2.2 SLA projections**

*The distinctive feature of this equatorial wave analysis compared to previous decompositions (e.g., Delcroix et al., 1994, BM9599) is the use of three-dimensional harmonic "wave" functions to be*

95   *projected onto the detrended SLA. Each wave function $F^q_{m,n,\varsigma,s}$ is the product of three components*

$$F^q_{m,n,\varsigma,s}(x,y,t) = \phi_q(y)G_{m,n,\varsigma}(x,t)H_s(x) \tag{1}$$

*namely, the meridional structure function $\phi_q(y)$ for meridional mode $q$, zonally-propagating harmonic functions $G_{m,n,\varsigma}(x,t)$, and a taper function $H_s(x)$. The parameters $m$, $n$, and $\varsigma$ indicate respectively the wavenumber, frequency, and phase of the function, while $s$ indicates the tapering location of the function as will be discussed in Section 2.2.3. (The coordinates $x$, $y$, and $t$ are the*

100   *zonal, meridional, and time coordinates respectively.) Each wave function is projected onto the SLA data between latitudes $15°S$ and $15°N$, to resolve the meridional structures of equatorial Kelvin waves and the first 5 meridional Rossby wave modes.*

**2.2.1 Meridional structure function**

*The meridional structure functions are derived from the eigenfunction solutions of the shallow-water*

105   *momentum and mass conservation equations, with the Coriolis parameter $f$ varying as a function of $y$ (e.g., Matsuno, 1966; Moore, 1968; Gill and Clarke, 1974). The form that these solutions take for pressure perturbations (and therefore for SLA) is*

$$\phi_{-1}(y) = \frac{1}{\sqrt{2}}\psi_0 \tag{2}$$

$$\phi_q(y) = \sqrt{\frac{q(q+1)}{2(2q+1)}} \left( \frac{\psi_{q+1}}{\sqrt{q+1}} + \frac{\psi_{q-1}}{\sqrt{q}} \right) \quad \text{for } q > 0 \tag{3}$$

*after Boulanger and Menkes (1995), with $\psi_q$ a solution (bounded as $y \to \pm\infty$) to the eigenvalue problem posed by the shallow-water equation for meridional velocity $v$*

$$\frac{\partial^2 v}{\partial y_*^2} + (2q+1-y_*^2)v = 0 \tag{4}$$

110   *with $\beta = \partial f/\partial y$ and non-dimensional $y_* = (\beta/c)^{1/2}y$. (The trivial solution $v = 0$ for (4) corresponds to $q = -1$, the Kelvin wave mode.) The value of the equatorial Kelvin wave phase speed $c$ was taken to be 2.5 m s$^{-1}$ in the Indian Ocean, based on earlier observational estimates for the first baroclinic wave phase speed (e.g., Nagura and McPhaden, 2012). In the Pacific Ocean, $c$ is taken to be 2.5 m s$^{-1}$ in the western part of the basin (e.g., Kessler and McPhaden, 1995; Hendon*

115 *et al., 1998), but east of 180° longitude this value is gradually decreased to 2.0 m s$^{-1}$ at the eastern*

*boundary due to the decrease in phase speed from the shoaling of the thermocline (e.g., Giese and*

*Harrison, 1990). More details on the derivation of these structure functions are available from other*

*sources (e.g., Matsuno, 1966; Boulanger and Menkes, 1995, , Sec. A1). The analytical form of $\psi_q$*

*for constant $\beta$ values can also be expressed in terms of Hermite polynomials (e.g., Matsuno, 1966);*

120 *in our case $\psi_q$ were solved for numerically, with $\beta$ values allowed to vary slightly as a function of*

*latitude.*

**2.2.2 Zonally-propagating harmonic functions**

*The zonally-propagating harmonic functions $G_{m,n,\varsigma}(x,t)$ take the form*

$$G_{m,n,1}(x,t) = \cos\left[2\pi\left(k_m x - f_n t\right)\right] \tag{5}$$

$$G_{m,n,2}(x,t) = \sin\left[2\pi\left(k_m x - f_n t\right)\right] \tag{6}$$

*with $\varsigma = 1$ indicating a cosine function (i.e. phase = 0°), and $\varsigma = 2$ indicating a sine function (i.e.*

125 *phase = 90°). Values of $k_m$ can vary in the range $k_m = 0, \pm 1/(x_L + \Delta x), \pm 2/(x_L + \Delta x), ...$ for*

*$|k_m| \leq 1/\Delta x$, with $x_L$ equal to the span of the ocean basin at the equator, and $\Delta x = 5°$. ($\Delta x$ is*

*chosen to be much larger than the actual zonal resolution of the data, as the long waves of interest*

*in this study generally have large zonal scales.) The wave functions are projected on to the low- and*

*high-passed SLA data separately as described in Section 2.1. For projections onto the low-passed*

130 *SLA data, values of $f_n$ can vary in the range $f_n = 0, 1/t_L, 2/t_L, ...$ for $f_n \leq 1/\Delta t$, with $t_L$ = 23*

*years (the time span of the entire dataset) and $\Delta t = 2/9$ years $\approx 81$ days (sufficient to resolve periods*

*longer than 162 days, or semiannual and lower frequencies). For the high-passed data, which are*

*projected in 2-year segments, $t_L = 2$ years and $f_n$ varies at intervals of $1/t_L$ in the frequency range*

*corresponding to periods of approximately 20–150 days (the intraseasonal frequency band), with*

135 *$\Delta t = 10$ days. As with $\Delta x$, $\Delta t$ can be longer than the actual time resolution of the data.*

**2.2.3 Taper functions**

*Lastly, the taper functions $H_s(x)$ are included to improve the representation of waves that are forced*

*and dissipate within the ocean basin. This is particularly important for equatorial long waves whose*

*zonal wavelength may be as long or longer than the width of the ocean basin. When propagating*

140 *waves change amplitude, a portion of the wave activity is aliased into adjacent wavenumbers and*

*frequencies; this presents a potential issue for correctly identifying the mode of low-wavenumber*

*equatorial long waves. The tapers help ensure that a low-wavenumber, eastward-propagating Kelvin*

*wave generated in the middle of the basin is identified as such. The taper functions (Figure 1) take the form*

$$H_s(x) = \begin{cases} 0, & x \leq x_s - \Delta x \\ \left(1 - \frac{x_s - x}{\Delta x}\right), & x_s - \Delta x < x < x_s \\ 1, & x \geq x_s \end{cases} \tag{7}$$

145   *The tapering location $x_s$ is varied within the range $x_s = 0, \Delta x, 2\Delta x, ....$ for $x_s \leq x_L$. (The analysis is carried out with two different $x$-coordinates: $x = 0$ at the western boundary and $x$ increases eastward, and $x = 0$ at eastern boundary and $x$ increases westward. The mode SLA obtained using the two coordinate systems are averaged at the end of the analysis; this approach is used to minimize any bias that would be introduced at either the western or eastern boundary due to the technique.)*

150 **2.2.4   Projecting wave functions for each mode**

*The wave functions $F^q_{m,n,\varsigma,s}$ for the Kelvin wave mode $q = 0$ and the first five meridional Rossby wave modes $q = 1, ..., 5$ are projected onto the detrended SLA. The wave functions for each mode $q$ consist of the corresponding meridional structure function $\phi_q$, a full range of possible taper functions $H_s$ as described in Section 2.2.3, and the propagating harmonic functions $G_{m,n,\varsigma}$ that have phase*

155   *speeds $f_n/k_m$ within the ranges given in Table 1. (Note: in the Pacific, where the input phase speed $c$ varies across the basin, $c_{min}$ is computed using $c = 2.0$ m s$^{-1}$, and $c_{max}$ is computed using $c = 2.5$ m s$^{-1}$. Hence the ranges specified in Table 1 will have some overlap, but the associated wave functions are still distinct because of their meridional structures.)*

  *As part of the signal for fast-propagating waves is aliased into the zero wavenumber band, $G_{m,n,\varsigma}$*

160   *functions for which $k_m = 0$ are also projected for frequency values $f_n < \Delta k |c_{\max}|$, with $\Delta k = 1/(x_L + \Delta x)$ the lowest-magnitude non-zero wavenumber resolved in this analysis, and $c_{\max}$ the maximum (magnitude) phase speed for that mode as given in Table 1.*

  *Prior to projecting, each wave function $F^q_{m,n,\varsigma,s}$ has the meridional mean and trend removed in the same way that the SLA data are processed. Then the projection of each wave function onto the*

165   *SLA data is carried out, point-by-point in the $x$, $y$, $t$ domain. If $\mathbf{d}$ is a vector consisting of all of the $x$, $y$, $t$ meridionally-detrended SLA data point values $\eta_d$, and $\mathbf{F}$ is a matrix with each column consisting of the $x$, $y$, $t$ values of a wave function $F^q_{m,n,\varsigma,s}$, then the projection value for column $\alpha$ of $\mathbf{F}$ (i.e., $\mathbf{F}_\alpha$) is given by*

$$p_\alpha = \left[ (\mathbf{F}_\alpha)^T \mathbf{F}_\alpha \right]^{-2} (\mathbf{F}_\alpha)^T \mathbf{d} \tag{8}$$

*with the first part of the right-hand side (containing the exponent -2) normalizing the projection. The*
170   *resulting scalars $p_\alpha$ are the elements of a vector $\mathbf{p}$ containing the projection values. Alternatively*
*(8) can be expressed in terms of a projection operator $\mathbf{P}$*

$$\mathbf{p} = \mathbf{Pd} \tag{9}$$

*We note that if all of the wave functions (i.e., the columns of $\mathbf{F}$) were independent, then (8) would*
*be equivalent to a linear regression of each wave function onto the SLA data, yielding coefficients*
*that could be used to compute the SLA associated with each mode. However, the wave functions are*
175   *not independent; furthermore, a linear regression applied directly to the SLA data weights the misfit*
*to points far from the equator (where wave amplitudes are negligible) just as heavily as points near*
*the equator where these waves constitute a large fraction of the SLA variance. Instead of regressing*
*the SLA data directly, a cost function of the misfit to projection values in $\mathbf{p}$ is minimized to find*
*optimal coefficients for the wave functions, as described in Section 2.3.*

180   ## 2.3   Least-squares deconvolution of wave function coefficients

*Given the projection values based on the SLA data, the next objective is to find a set of "true"*
*wave function coefficients $\mathbf{m}$ that can most accurately account for the observation-based projection*
*values in $\mathbf{p}$, within desired parameters. To achieve this, we seek to solve the system $\mathbf{Fm} = \mathbf{d}$ for the*
*coefficients $\mathbf{m}$, such that the following cost function is minimized*

$$L = [\mathbf{DP}\left(\mathbf{Fm}\right) - \mathbf{Dp}]^T [\mathbf{DP}\left(\mathbf{Fm}\right) - \mathbf{Dp}] + [w\left(\mathbf{P}\left(\mathbf{Fm}\right) - \mathbf{p}\right)]^T [w\left(\mathbf{P}\left(\mathbf{Fm}\right) - \mathbf{p}\right)] + \left[\left(\mathbf{\Lambda m}\right)^T \mathbf{m}\right] \tag{10}$$

185   *There are three parts to this cost function. The first part minimizes the misfit to the change in the*
*projection values along the waveguide $\mathbf{Dp}$, the second part minimizes the misfit to the projection*
*values $\mathbf{p}$, and the third part establishes a preference for values of $\mathbf{m}$ that are not too large. The*
*matrix $\mathbf{D}$ is the finite difference operator for $s > 1$ projections, and the identity operator for $s = 1$*
*projections (i.e., the finite difference between zero and the $s = 1$ projection value). The scalar value*
190   *$w$ and the diagonal matrix $\mathbf{\Lambda}$ are both adjustable parameters that set the relative weighting of the*
*parts of the cost function. A number of possible combinations of these parameters were tested to*
*determine which values generated the solution that explains the most SLA variance. The optimal*
*value chosen for $w$ is 0.2; the diagonal of $\mathbf{\Lambda}$ has values of 2 for entries corresponding to $s = 1$*
*wave functions (i.e., the functions that span the entire ocean basin). For the $s > 1$ wave functions,*
195   *values of 20 are used in the Indian Ocean, and 5 in the Pacific Ocean. The different values in the*
*two basins are likely needed to suppress high-wavenumber variability in the Indian Ocean that is*

*not well described by the low-wavenumber limits of the equatorial wave dispersion relation (e.g.*
*Matsuno, 1966).*

   *The formula for the coefficients* $\mathbf{m}$ *that minimizes the cost function (10) is*

$$\mathbf{m} = \left[ (\mathbf{DPF})^T \mathbf{DFP} + (w\mathbf{PF})^T w\mathbf{PF} + \mathbf{\Lambda} \right]^{-1} \left[ (\mathbf{DPF})^T \mathbf{Dp} + (w\mathbf{PF})^T w\mathbf{p} \right] \tag{11}$$

*Once the coefficients* $\mathbf{m}$ *are obtained, then the contribution to SLA from wave functions associated*
*with each mode q can be summed up to yield the contribution to SLA from that mode* $\eta_q$ *(the mode*
*SLA):*

$$\eta_q(x,y,t) = \sum_q \mathbf{m}^q_{m,n,c,s} F^q_{m,n,\varsigma,s}(x,y,t) \tag{12}$$

*with* $\mathbf{m}^q_{m,n,c,s}$ *the (scalar) coefficient from* $\mathbf{m}$ *that corresponds to the wave function* $F^q_{m,n,\varsigma,s}$. *(Note:*
*the meridional structure functions within* $F^q_{m,n,\varsigma,s}$ *have their original meridional means and trends*
*added back to them before the mode SLA is calculated.)*

**3   Validation and characteristics of the mode SLA**

*To assess how accurately the mode SLA represents the true contribution of each meridional mode to*
*the SLA field, two correlation metrics are computed. The first correlates the temporal variation of*
*SLA associated with each mode to that of the original, meridionally-detrended SLA* $r(\eta_q, \eta_d)$. *Robust*
*values of this correlation at latitudes where the given wave mode q is known to make a leading-order*
*contribution to SLA variability confirm that the sign and meridional structure of the wave mode is*
*well represented. The other metric is the correlation of the temporal variation of mode SLA to the*
*residual SLA* $\eta_r$, *which is* $\eta_d$ *with the sum of the mode SLA from all of the projected modes removed*

$$\eta_r(x,y,t) = \eta_d - \sum_q \eta_q \tag{13}$$

*The temporal correlation* $r(\eta_q, \eta_r)$ *assesses whether the mode SLA is accurately representing the*
*amplitude of the waves. If (1) the SLA signal actually associated with a given mode q is well repre-*
*sented by the mode SLA, and (2) other signals are not significantly correlated with variations due*
*to mode q, then* $r(\eta_q, \eta_r)$ *should be insignificant especially at latitudes where the wave amplitude*
*peaks. It is important to note that condition (2) may not be met, as other signals that were not pro-*
*jected in this method may have SLA variations that are correlated with the projected modes (e.g.,*
*higher-order meridional and baroclinic wave modes, MJO-related variations that do not project onto*
*an equatorial wave mode but are coincident with mode forcing). However, it is expected that for the*

*most part SLA signals near the equator will be explained either by the long (zonal) wavelength equa-torial waves accounted for in this analysis, or shorter wavelength signals that are uncorrelated with equatorial long wave activity.*

225     *These correlation analyses were used to tune the variable parameters $w$ and $\mathbf{\Lambda}$. The correlation values for the optimal parameters given in Section 2.3 are mapped for the equatorial Indian Ocean in Figures 2 and 3. For the Kelvin wave mode ($q = -1$), the correlation $r(\eta_q, \eta_d)$ is highest along the equator (Fig. 2a), while for the Rossby wave mode $q = 1$ the correlation is highest near the latitude of the wave's peaks, 3°–4° north and south of the equator (Fig. 2b). The SLA fields were*

230 *also bandpassed for intraseasonal frequencies (20–150 day periods) to focus on freely-propagating waves that are forced by and interact with MJO winds (e.g., Han et al., 2001). Compared to lower-frequency equatorial waves, it is easier to resolve the zonal wavenumbers of intraseasonal equatorial waves, and therefore their propagation characteristics. The correlations at these frequencies (Fig. 2c,d) are slightly lower in magnitude than for the non-bandpassed mode SLA, but still peak at the*

235 *expected latitudes, confirming that $c = 2.5\ m\ s^{-1}$ is a suitable choice for defining the meridional structure of these lowest-mode waves. Moreover, the non-bandpassed $q = 1$ correlations have some maxima slightly equatorward of their expected latitudes (Fig. 2b), but the bandpassed correlation maxima track the expected latitudes for $c = 2.5\ m\ s^{-1}$ nearly exactly (Fig. 2d). The 2.5 $m\ s^{-1}$ phase speed is consistent with the Argo-based 1st baroclinic mode phase speed estimate by Nagura and*

240 *McPhaden (2012); according to the estimate by Drushka et al. (2010), also based on Argo data, this value would lie between the 1st and 2nd baroclinic mode Kelvin wave phase speeds (2.8 and 1.8 m $s^{-1}$ respectively). As Drushka et al. (2010) and others have found, the first and second baroclinic modes make substantial contributions to intraseasonal wave activity in the equatorial Indian Ocean, so the 2.5 $m\ s^{-1}$ phase speed value may represent waves that contain both baroclinic modes.*

245     *The Kelvin and $q = 1$ Rossby mode SLA correlations with the residual SLA (Figure 3) suggest that a consideration of timescales is important when tuning the parameters $w$ and $\mathbf{\Lambda}$ to optimize the accuracy of the mode SLA. The non-bandpassed Kelvin wave mode SLA correlation in particular (Fig. 3a) indicate banded features in the northwest Indian Ocean that have little to do with the Kelvin wave structure; it turns out that these are mostly a result of the annual and semiannual*

250 *signals associated with monsoonal wind reversals. By focusing on correlations with intraseasonal mode SLA (Fig. 3c,d) it is possible to obtain mode SLA values that are not significantly correlated (or only weakly correlated) with the residual SLA near the peak amplitude axes of the waves. The intraseasonal Kelvin wave SLA is slightly positively correlated with the residual SLA along most of the equator, and negatively correlated just off the equator (Fig. 3c); on balance this suggests that*

255 *there is little bias in the mode SLA estimations of the waves' amplitude. Weak positive or insignificant correlations are also found along the $q = 1$ Rossby wave peak amplitude axes, with negative SLA correlations elsewhere. The exception is north of the equator and west of the longitudes of India/Sri Lanka, where correlations are almost uniformly negative. The negative correlations would imply*

*that the amplitude of the $q = 1$ Rossby waves are being overestimated in this region, though these*
260 *inaccuracies in the mode SLA might be explained by interference from the landmasses of Sri Lanka and India.*

**4 Mode SLA representations of the equatorial ocean**

*The mode SLA produced by this method has several desirable features compared to earlier tech-niques used to describe equatorial wave activity. These features include zonal wave propagation*
265 *that is constrained (though weakly) to within phase speed ranges predicted by theory, and a more active intraseasonal wave field than BM9599 that still has sufficiently long wavelengths ($> 10°$ lon-gitude) to fall within the essentially non-dispersive range of long equatorial wave propagation (e.g. Matsuno, 1966). Figure 4 shows a snapshot of the progression of Kelvin and mode 1 Rossby waves across the Indian Ocean, as tracked by mode SLA during the active El Niño and positive IOD year*
270 *1997. During May-July 1997, the period portrayed in Fig. 4, westerly winds along the equator tran-sitioned to anomalously strong easterly winds, forcing an upwelling Kelvin wave along the equator as indicated by negative values of mode SLA (Fig. 4g,h). The reflection of a downwelling Kelvin wave at the eastern boundary (Fig. 4e) is seen as a mode 1 Rossby wave that propagates westward from the boundary to the center of the Indian Ocean (Fig. 4j-l). Furthermore, the propagation char-*
275 *acteristics of these waves can be much more readily observed in the mode SLA compared to the raw SLA data.*

*Hovmöller diagrams of mode SLA amplitudes during 1993–1997 in the equatorial Indian (Figure 5) and Pacific (Figure 6) oceans illustrate wave variations at intraseasonal, annual, and interannual timescales. In the Indian Ocean, Kelvin wave activity largely reflects the annual cycle forced by wind*
280 *reversals between the northeast and southwest phases of the South Asian monsoon (Fig. 5a); these waves are then generally reflected at the eastern boundary as $q = 1$ Rossby waves (Fig. 5b). The Pacific Ocean Hovmöller diagrams indicate a large amount of interannual as well as intraseasonal wave activity; the dominant signals during the 1997 El Niño (and during the weaker 1994 El Niño before it) are the downwelling Kelvin waves generated, then reflected as downwelling $q = 1$ Rossby*
285 *waves (Fig. 6a,b). Notably, in the $q = 3$ mode, the Indian and Pacific Ocean basins have overwhelm-ingly downwelling and upwelling waves respectively during the 1997 El Niño and positive IOD event (Fig. 5d, 6d). This anomaly is not shown consistently in the $q = 1$ modes for each basin (Fig. 5b, 6b), suggesting that the third meridional Rossby wave mode may be as important as the first mode in determining the evolution of the extreme 1997 event.*
290 *The amplitudes of the Pacific mode SLA for Kelvin and $q = 1$ Rossby waves are also plotted to illustrate the reflection of equatorial waves at the boundaries (Figure 7). This figure takes the same form and spans a similar time range to Figure 6 in Boulanger and Menkes (1999). In comparing Fig. 7 with the Boulanger and Menkes (1999) figure, it can be seen that the method used here highlights*

*the higher-frequency intraseasonal wave activity more clearly, such as the downwelling Kelvin waves*
*at the end of 1996 and in early 1997 that initiated the El Niño state. By contrast, the Boulanger and*
*Menkes (1999) wave coefficients contain some non-propagating structure not present in our Figure*
*7, such as the difference in sign between the western and eastern Pacific (for both Kelvin and Rossby*
*wave coefficients) at the height of the 1997 El Niño. It makes sense that incorporating the zonal*
*propagation of equatorial waves into an SLA projection method would be clearest at intraseasonal*
*frequencies. At annual and longer timescales, Kelvin and lowest-mode equatorial Rossby waves*
*mostly exist as standing features with meridional structures, as the first baroclinic mode of these*
*waves may be in phase across most or all of the basin. But at intraseasonal timescales, the zonal*
*phase propagation is an essential characteristic that may aid in the separation of meridional (and*
*potentially baroclinic) modes of wave activity.*

**5   Conclusions**

The harmonic projection and least-squares method outlined here produces a measure of *equatorial*
wave activity that can be *derived* directly *from* satellite observations of SLA. The method *projects*
*onto the data SLA field a set of wave functions that resemble zonally-propagating waveforms, then*
*minimizes a cost function to deconvolve the coefficients of the non-orthogonal wave functions. The*
*result of these calculations is a mode SLA field for equatorial Kelvin waves and the first lowest 5*
*meridional Rossby wave modes. Correlations of the mode SLA fields with the data SLA and residual*
*SLA fields help to tune the variable parameters in the calculation, and confirm that $c = 2.5 \ m \ s^{-1}$*
*is an accurate choice for the Indian Ocean (1st baroclinic) Kelvin wave phase speed. Compared to*
*the earlier SLA decomposition method used by BM9599 based solely on meridional profiles of the*
*waves, the SLA field generated by this method contains more variability at intraseasonal frequencies,*
*and downplays the role of non-propagating meridional structure in the SLA field. Our method* also
allows for some variation in the phase speed of the waves, so a comparison of *the wave amplitudes*
*derived from mode SLA* with the results of linear wind–forced models of equatorial waves (e.g.,
Yu and McPhaden, 1999; Nagura and McPhaden, 2010) may be useful in studying some weakly
nonlinear aspects of Kelvin waves.

*One caveat for the use of the mode SLA is that the method favors waveforms that vary at simi-*
*larly large scales across the ocean basin; near the boundaries or local bathymetric features, it may*
*be desirable to resolve smaller scales more accurately. While the reflection of equatorial waves is*
*quite apparent in our wave amplitude plots (Fig. 5–7), there are cases where an abrupt change in*
*sign occurs from the incoming to the reflected wave. An increase in resolution near the boundaries*
*by incorporating more taper functions in the basis set used could be one way to improve represen-*
*tations of wave reflections. An advantage of the methodological framework described in Section 2*
*is that like most linear inverse methods it is very adaptable; any signal that has a fairly consistent*

*structure in one (or two or three) dimensions and propagates in a favored direction could in theory*
330 *be represented by wave functions of the form given in equation (1). The accurate recovery of the*
*signal using this framework depends on the choice of correct wave function structure and tuning of*
*the cost function against validation metrics.*

*Acknowledgements.* Andrew Delman (ASD) was supported by a NASA Earth and Space Science Fellowship, grant number NNX13AM93H. Janet Sprintall (JS) and Julie McClean (JLM) were supported by NASA
335 award number NNX13AO38G. Lynne Talley (LDT), JLM, and ASD were also supported by NSF grant OCE–0927650. The altimeter products were produced by Ssalto/Duacs and distributed by Aviso, with support from CNES (http://www.aviso.oceanobs.com/duacs/). Computations were carried out on the Geyser cluster within the Yellowstone computing environment hosted by the National Center for Atmospheric Research (NCAR). *We would also like to thank two reviewers for their comments which have helped in improving this manuscript.*

340 **References**

Boulanger, J.-P. and Menkes, C.: Propagation and reflection of long equatorial waves in the Pacific Ocean during the 1992–1993 El Niño, J. Geophys. Res., 100, 25,041–25,059, 1995.

Boulanger, J.-P. and Menkes, C.: Long equatorial wave reflection in the Pacific Ocean from TOPEX/POSEIDON data during the 1992–1998 period, Clim. Dyn., 15, 205–225, 1999.

345 Cane, M. and Sarachik, E.: The response of a linear baroclinic equatorial ocean to periodic forcing, J. Mar. Res., 39, 651–693, 1981.

Delcroix, T., Boulanger, J.-P., Masia, F., and Menkes, C.: Geosat–derived sea level and surface current anomalies in the equatorial Pacific during the 1986–1989 El Niño and La Niña, J. Geophys. Res., 99, 25,093–25,107, 1994.

350 Drushka, K., Sprintall, J., Gille, S. T., and Brodjonegoro, I.: Vertical structure of Kelvin waves in the Indonesian Throughflow exit passages, J. Phys. Oceanogr., 40, 1965–1987, 2010.

Ducet, N., Traon, P. Y. L., and Reverdin, G.: Global high–resolution mapping of ocean circulation from TOPEX/Poseidon and ERS–1 and –2, J. Geophys. Res., 105, 19,477–19,498, 2000.

Fu, L.-L., Fukumori, I., and Miller, R. N.: Fitting dynamic models to the Geosat sea level observations in the
355 tropical Pacific ocean. Part II: a linear, wind–driven model, J. Phys. Oceanogr., 23, 2162–2181, 1993.

Giese, B. S. and Harrison, D. E.: Aspects of the Kelvin wave response to episodic wind forcing, J. Geophys. Res., 95, 7289–7312, 1990.

Gill, A. E. and Clarke, A. J.: Wind–induced upwelling, coastal currents, and sea–level changes, Deep-Sea Res., 21, 325–345, 1974.

360 Han, W., Lawrence, D. M., and Webster, P. J.: Dynamical response of equatorial Indian Ocean to intraseasonal winds: zonal flow, Geophys. Res. Lett., 28, 4215–4218, 2001.

Hendon, H. H., Liebmann, B., and Glick, J. D.: Oceanic Kelvin waves and the Madden–Julian oscillation, J. Atmos. Sci., 55, 88–101, 1998.

Kessler, W. S. and McPhaden, M. J.: Oceanic equatorial waves and the 1991–93 El Niño, J. Climate, 8, 1757–
365 1774, 1995.

Long, R. B. and Thacker, W. C.: Data assimilation into a numerical equatorial ocean model. I. The model and the assimilation algorithm, Dyn. Atmos. Oceans, 13, 379–412, 1989a.

Long, R. B. and Thacker, W. C.: Data assimilation into a numerical equatorial ocean model. II. Assimilation experiments, Dyn. Atmos. Oceans, 13, 413–439, 1989b.

370 Matsuno, T.: Quasi–geostrophic motions in the equatorial area, J. Met. Soc. Japan, 44, 25–43, 1966.

Miller, R. N. and Cane, M. A.: A Kalman filter analysis of sea level height in the tropical Pacific, J. Phys. Oceanogr., 19, 773–790, 1989.

Moore, D. W.: Planetary-gravity waves in an equatorial ocean, Ph.D. thesis, Harvard University.

Nagura, M. and McPhaden, M. J.: Dynamics of zonal current variations associated with the Indian Ocean dipole,
375 J. Geophys. Res., 115, C11026, doi:10.1029/2010JC006423., 2010.

Nagura, M. and McPhaden, M. J.: The dynamics of wind-driven intraseasonal variability in the equatorial Indian Ocean, J. Geophys. Res., 117, C02001, doi:10.1029/2011JC007405., 2012.

Susanto, R. D., Zheng, Q., and Yan, X.-H.: Complex singular value decomposition analysis of equatorial waves in the Pacific observed by TOPEX/Poseidon altimeter, J. Atmos. Oceanic Technol., 15, 764–774, 1998.

380 Thacker, W. C. and Long, R. B.: Fitting dynamics to data, J. Geophys. Res., 93, 1227–1240, 1988.

Yu, X. and McPhaden, M. J.: Seasonal variability in the equatorial Pacific, J. Phys. Oceanogr., 29, 925–947, 1999.

Yuan, D. and Liu, H.: Long-wave dynamics of sea level variations during Indian Ocean Dipole events, J. Phys. Oceanogr., 39, 1115–1132, 2009.

385 Yuan, D., Rienecker, M. M., and Schopf, P. S.: Long-wave dynamics of the interannual variability in a numerical hindcast of the equatorial Pacific Ocean circulation during the 1990s, J. Geophys. Res., 109, C05019, 2004.

[Figure]

**Figure 1.** Schematic illustrating the use of taper functions. (a) Profile of a non–tapered harmonic function $G_{m,n,\varsigma}$ in $x$ and $t$. (b) Profile of the same harmonic function modified or "forced" by a taper function, $G_{m,n,\varsigma}H_s$, with tapering location $x = x_s$ and a tapering window of $\Delta x$. (c) Profile of a harmonic function that is "forced" and "dissipated" by two taper functions $G_{m,n,\varsigma}(H_s - H_{s'})$, with tapering locations of $x = x_s$ and $x = x_{s'}$ respectively.

**Table 1.** Phase speed ranges for zonally-propagating harmonic functions $G_{m,n,\varsigma}^q$ associated with mode $q$. Positive (negative) values indicate eastward (westward) propagating harmonic functions. For each mode, $c_{\min}$ and $c_{\max}$ are the minimum- and maximum-magnitude speeds in the range, regardless of direction, i.e., $c_{\min} \leq f_n/k_m \leq c_{\max}$ for Kelvin waves ($q = -1$) and $c_{\max} \leq f_n/k_m \leq c_{\min}$ for Rossby waves ($q > 0$).

| Mode $q$ | $c_{\min}$ | Theoretical phase speed | $c_{\max}$ |
|:---:|:---:|:---:|:---:|
| -1 | +0.6$c$ | +$c$ | +1.4$c$ |
| 1 | -$c$/4 | -$c$/3 | -$c$/2 |
| 2 | -$c$/6 | -$c$/5 | -$c$/4 |
| 3 | -$c$/8 | -$c$/7 | -$c$/6 |
| 4 | -$c$/10 | -$c$/9 | -$c$/8 |
| 5 | -$c$/12 | -$c$/11 | -$c$/10 |

[Figure]

**Figure 2.** (a) Temporal correlation $r(\eta_q, \eta_d)$ of the Kelvin wave mode SLA ($q = -1$) with the meridionally detrended SLA from the AVISO $1/4°$ daily gridded product, for the time range 1993–2015. A zonal low-pass filter is applied to $\eta_d$ prior to computing the correlations, to remove wavelengths shorter than $10°$. Only correlation coefficients surpassing the 95% confidence threshold for significance are shaded. (b) Same as (a), but for the correlation of the first meridional mode Rossby wave SLA ($q = 1$) with the meridionally detrended AVISO data. The horizontal dashed line indicates the latitude at which the peak amplitudes of the $q = 1$ Rossby wave occur. (c)-(d) Same as (a)-(b) respectively, but the correlations are computed after bandpassing $\eta_q$ and $\eta_d$ for intraseasonal frequencies in the 20–150 day period range.

[Figure]

**Figure 3.** (a) Temporal correlation $r(\eta_q, \eta_r)$ of the Kelvin wave mode SLA ($q = -1$) with the residual SLA unexplained by the Kelvin and Rossby wave mode SLA, for the time range 1993–2015. A zonal low-pass filter is applied to $\eta_r$ prior to computing the correlations, to remove wavelengths shorter than $10°$. Only correlation coefficients surpassing the 95% confidence threshold for significance are shaded. (b) Same as (a), but for the correlation of the first meridional mode Rossby wave SLA ($q = 1$) with the residual SLA. The horizontal dashed line indicates the latitude at which the peak amplitudes of the $q = 1$ Rossby wave occur. (c)-(d) Same as (a)-(b) respectively, but the correlations are computed after bandpassing $\eta_q$ and $\eta_r$ for intraseasonal frequencies in the 20–150 day period range.

[Figure]

**Figure 4.** (a)-(d) Maps of the sea level anomaly in the Indian Ocean from AVISO, for a series of dates in May-July 1997. (e)-(h) Maps of the Kelvin wave mode SLA ($\eta_q$ with $q = -1$) for the same dates. (i)-(l) Maps of the first meridional Rossby wave mode SLA ($\eta_q$ with $q = 1$) for the same dates.

[Figure]

**Figure 5.** (a) Kelvin wave $q = -1$ mode SLA peak amplitudes (i.e., the mode SLA along the latitude where the wave signal peaks) in the Indian Ocean, plotted from 1993 to 1997. (b) Rossby wave $q = 1$ mode SLA peak amplitudes; in this plot the $x$-axis is reversed in direction to highlight wave reflection at the eastern boundary. (c) Rossby wave $q = 2$ mode SLA peak amplitudes. (d) Rossby wave $q = 3$ mode SLA peak amplitudes.

[Figure]

**Figure 6.** Same as Figure 5, but in the equatorial Pacific Ocean.

[Figure]

**Figure 7.** (a) Kelvin wave $q = -1$ mode SLA amplitudes in the equatorial Pacific Ocean, plotted from 1993 to 1997 and normalized in the same way as in Boulanger and Menkes (1999) (1 BM99 unit = 0.32 cm wave amplitude). (b) Rossby wave $q = 1$ mode SLA amplitudes with the $x$-axis direction reversed; 1 BM99 unit = 0.28 cm wave amplitude. (c) Same as (a). The layout of (a)-(c) highlights the reflection of waves at the eastern and western boundaries; compare to Fig. 6 in Boulanger and Menkes (1999).